# EF-3DGS: Event-Aided Free-Trajectory 3D Gaussian Splatting

Bohao Liao[1][*]    Wei Zhai[1][†]    Zengyu Wan[1]    Zhixin Cheng[1]
Wenfei Yang[1]    Yang Cao[1]    Tianzhu Zhang[1]    Zheng-Jun Zha[1]

[1] University of Science and Technology of China

{liaobh, wanzengy, chengzhixin}@mail.ustc.edu.cn,
{wzhai056, yangwf, forrest, tzzhang, zhazj}@ustc.edu.cn

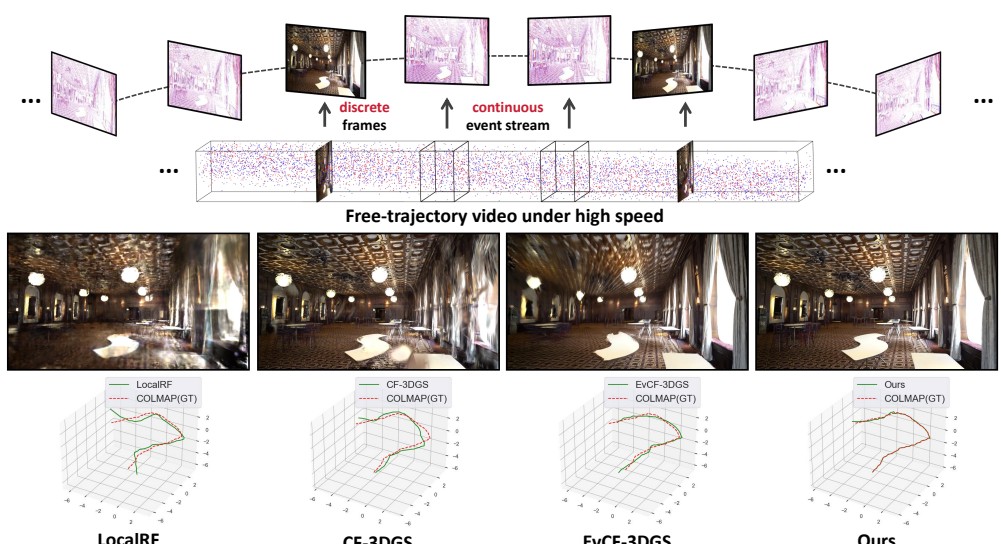

Figure 1: Free-trajectory 3DGS under high speed. (**Top**) The overall paradigm. The colored dots in the top row represent the event data (red: positive, blue: negative). We leverage continuous event streams to aid discrete video frames captured along free trajectories in high-speed scenarios, jointly optimizing camera poses and reconstructing the 3DGS. Our method surpasses current state-of-the-art methods in terms of both rendered results (**middle**) and pose estimation (**bottom**).

## Abstract

Scene reconstruction from casually captured videos has wide real-world applications. Despite recent progress, existing methods relying on traditional cameras tend to fail in high-speed scenarios due to insufficient observations and inaccurate pose estimation. Event cameras, inspired by biological vision, record pixel-wise intensity changes asynchronously with high temporal resolution and low latency, providing valuable scene and motion information in blind inter-frame intervals. In this paper, we introduce the event cameras to aid scene construction from a casually captured video for the first time, and propose Event-Aided Free-Trajectory 3DGS, called **EF-3DGS**, which seamlessly integrates the advantages of event cameras into 3DGS through three key components. First, we leverage the Event

---

[*]First author

[†]Corresponding author

39th Conference on Neural Information Processing Systems (NeurIPS 2025).

Generation Model (EGM) to fuse events and frames, enabling continuous supervision between discrete frames. Second, we extract motion information through Contrast Maximization (CMax) of warped events, which calibrates camera poses and provides gradient-domain constraints for 3DGS. Third, to address the absence of color information in events, we combine photometric bundle adjustment (PBA) with a Fixed-GS training strategy that separates structure and color optimization, effectively ensuring color consistency across different views. We evaluate our method on the public Tanks and Temples benchmark and a newly collected real-world dataset, RealEv-DAVIS. Our method achieves up to 3dB higher PSNR and 40% lower Absolute Trajectory Error (ATE) compared to state-of-the-art methods under challenging high-speed scenarios.

# 1   Introduction

In recent years, Neural Radiance Fields (NeRF) [1, 2, 3] and 3D Gaussian splatting (3DGS) [4, 5] have made significant progress in novel view synthesis tasks. Given a set of posed images of the same scene, they optimize an implicit or explicit scene representation using volume rendering. While subsequent methods [2, 3, 6, 7] excel with posed images, reconstructing scenes from videos with free camera trajectories remains challenging despite its applications in VR/AR, video stabilization, and mapping. To tackle this challenging task, several efforts have been made.

Accurate pose estimation is often difficult to obtain in free-trajectory scenarios, which directly impacts the quality of scene reconstruction. One line of work draws inspiration from Simultaneous Localization and Mapping (SLAM). They [8, 5, 9] follow its optimization paradigm, progressively optimizing camera trajectories and alternating between camera pose and scene refinement. Another line of work [10, 9, 11, 5, 12] explores incorporating additional geometric or motion priors such as depth estimation [13, 14] or optical flow [15] to establish constraints beyond photometric rendering loss. While these methods can render photo-realistic images in typical free-trajectory scenarios, both their rendering quality and pose estimation accuracy degrade significantly in high-speed scenarios (or equivalently low-frame-rate scenarios) as shown in Fig. 1. Such high-speed scenarios have essential applications such as autonomous driving and First-Person View (FPV) exploration.

The performance degradation of prior methods can be attributed to two primary factors. First, the limited number of camera observations leads to an under-constrained scene reconstruction problem. This can cause the scene representation to converge to a trivial solution [16, 17, 12], where the model overfits to the training views without capturing the correct underlying geometry structure. Second, the substantial discrepancies between consecutive frames, resulting in diminished overlapping regions, violate the implicit assumption of continuous motion between adjacent frames, which is leveraged by previous methods. Moreover, geometric and motion priors like optical flow and feature matching become unreliable in such scenarios. These significant violations greatly exacerbate the ill-posedness of the joint optimization of scene and camera poses.

Event camera is a bio-inspired image sensor that asynchronously records per-pixel brightness changes, offering advantages such as high temporal resolution, high dynamic range, and no motion blur [18, 19, 20, 21, 22, 23]. The brightness information recorded in the event stream can effectively complement the missing scene information between consecutive frames. Moreover, the event data naturally encodes the motion information of the scene [24, 25, 26], containing rich motion cues. These properties make event cameras well-suited for scene reconstruction tasks in high-speed and free-trajectory scenarios. However, seamlessly integrating the aforementioned benefits of event cameras is nontrivial. First, 3DGS renders absolute pixel brightness, which aligns with image data. Event cameras, however, record sparse differential brightness changes. Directly integrating the differential operations into 3DGS may amplify noise and lead to ill-conditioned optimization problems with high sensitivity to parameter initialization and perturbations. Second, event cameras encode motion through continuous spatio-temporal trajectories of events. In contrast, frame-based data inherently discretizes continuous motion, forcing traditional methods to rely on correspondence matching, which fails in high-speed scenarios with large inter-frame displacements. These fundamental challenges require carefully designed method that bridges the gap between the event data and 3DGS optimization.

In this work, we propose Event-Aided Free-Trajectory 3DGS, dubbed EF-3DGS, a framework that integrates event data into the scene optimization process to fully leverage its high temporal resolution property. Our approach comprises three key components: (1) In the Event Generation Model (EGM),

we introduce an event-based re-render loss, which extends the 3DGS optimization to the continuous event stream. This allows us to utilize the brightness cues encoded in the event stream between adjacent frames, providing rich supervisory signals to alleviate the insufficient sparse view issues. (2) In the Linear Event Generation Model (LEGM), regarding the pose estimation challenge, we introduce the CMax [27] framework to exploit the spatio-temporal correlations of events. We obtain the motion field by leveraging the pseudo-depth from 3DGS rendering and the relative camera motion between consecutive frames. We then warp the events triggered by the same edge along the motion trajectories to maximize the sharpness of the image of warped events (IWE), thereby estimating the motion that best matches the current spatio-temporal event patterns. Furthermore, through the LEGM [28, 29], we establish a connection between motion and brightness changes. This allows us to constrain the 3DGS in the gradient domain using the IWE. (3) As most event data primarily records scene brightness changes, lacking color information, we introduce photometric bundle adjustment (PBA) and a Fixed-GS strategy to address this. PBA recovers color by optimizing reprojection errors onto RGB frames, while Fixed-GS enables separate optimization of scene structure and color.

Our main contributions are summarized as follows:

- We introduce event cameras into the task of free-trajectory scene reconstruction for the first time. Its advantage of high temporal resolution and low latency showcases the potential of event data for scene reconstruction tasks in challenging scenarios.

- We derive our method from the underlying imaging principles of event cameras and design the corresponding loss functions that mine the motion and brightness information encoded in event data and seamlessly integrate them into the 3DGS optimization.

- Experiments on both public benchmarks and real-world datasets demonstrate that our method significantly outperforms existing state-of-the-art approaches in terms of both rendering quality and trajectory estimation accuracy.

## 2 Related Works

**Joint Pose and Scene Optimization.** The research community has recently focused on developing methods [12, 8, 30, 31, 10, 32, 5] that can be optimized without requiring precomputed camera poses. A line of work has focused on improving the stability of the optimization process. GARF [31] and BARF [32] both find that the high-frequency position encoding is prone to local minima and try to improve it. For example, GARF [31] proposes using Gaussian activation to replace the sinusoidal position encoding. Another line of work has investigated incorporating additional constraints to make the problem more tractable. LocalRF [8] leverages the prior assumption of continuous motion between adjacent frames and progressively adds and optimizes camera poses. More recent approaches [10, 8, 5] leverage pre-trained networks, *i.e.*, monocular depth estimation and optical flow estimation. Exploiting 3DGS's explicit representation, CF-3DGS [5] directly back-projects Gaussian points using depth maps. While the aforementioned methods have made notable progress, they have yet to fully address the challenges posed by high-speed scenarios or rely on a good pose initialization. Our approach addresses these issues by leveraging motion and brightness cues from event streams.

**Event-Based Novel View Synthesis.** Recent works have explored the integration of event cameras [33, 34, 35] into the NeRF or 3DGS framework. Early approaches, such as E-NeRF [36] and EventNeRF [37], utilize event-based generative models, minimizing the difference between the rendered brightness changes and observed brightness changes. Building upon this, Robust e-NeRF [38] incorporates a more realistic imaging model into the event-based framework, accounting for factors like refractory periods and noise. Beyond event-based NeRF, efforts have also been made to integrate event data into image-based methods. For instance, E2NeRF [39] and EvDeblurNeRF [40] leverage the Event Double Integral (EDI) [41] model to address the deblurring problem, while DE-NeRF [42] and EvDNeRF [43] leverage the high temporal resolution property of event cameras to capture fast-moving elements in dynamic scene. More recently, Event-3DGS [44] and EaDeblur-GS [45] have extended previous approaches to 3D Gaussian Splatting, achieving superior rendering quality and real-time performance. A key distinction of our work is that, unlike the prior methods that rely on accurate precomputed poses, we target free-trajectory scenarios, jointly optimizing for both the camera poses and the scene representation. Furthermore, while previous works have been limited to simulated and simple environments, we evaluate our approach in large-scale outdoor scenarios with complex motions and lighting conditions.

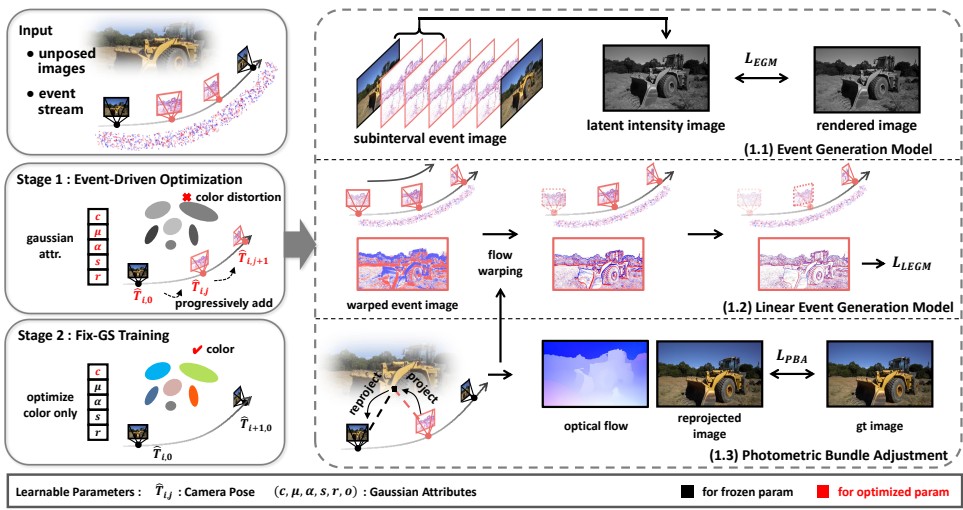

Figure 2: Method overview. The inputs are video frames and event stream. In the first stage, we progressively add new event images, leveraging the events and most recent frame to establish the event-driven optimization. In the second stage, we adopt the Fixed-GS strategy to mitigate the color distortion of 3DGS. The details of $\mathcal{L}_{LEGM}$ and CMax framework are shown in Fig. 3.

## 3 Preliminary

3DGS [4] parametrizes the 3D scene as a set of 3D gaussians $\{G_k\}_{k=1}^{K}$ that carry the geometric and appearance information. Each 3D Gaussian is characterized by several learnable properties, including its center position $\mu \in \mathbb{R}^3$, opacity $\alpha \in [0, 1]$, spherical harmonics (SH) features $\mathbf{f}_k \in \mathbb{R}^{3 \times 16}$ for view-dependent color $c \in \mathbb{R}^3$, rotation matrix $R \in \mathbb{R}^{3 \times 3}$ (stored in quaternion form), scale factor $s \in \mathbb{R}^3$. The shape of each Gaussian is defined by the covariance matrix $\mathbf{\Sigma}$ and the center (mean) point $\mu$, $G(x) = \exp(-\frac{1}{2}(x - \mu)^T \mathbf{\Sigma}^{-1}(x - \mu))$. During rendering, a tile-based rasterizer is applied to enable fast sorting and $\alpha$-blending. The color of each pixel is calculated via blending N ordered overlapping points:

$$C(\mathbf{r}) = \sum_{i=1}^{N} c_i \alpha_i \prod_{j=1}^{i-1}(1 - \alpha_j),\tag{1}$$

where $c_i$ is calculated from spherical harmonics and view direction, $\alpha_i$ is the multiplication of opacity and the transformed 2D Gaussian and $\mathbf{r}$ denotes the image pixel. With the forward rendering procedure, we can optimize 3DGS by minimizing a weighted combination loss of $\mathcal{L}_1$ and $\mathcal{L}_{D-SSIM}$ between observation and rendered pixels: $\mathcal{L}_{color} = (1 - \lambda)\mathcal{L}_1(\hat{I}, I) + \lambda \mathcal{L}_{D-SSIM}(\hat{I}, I)$, where $\lambda$ is balancing weight which is set to 0.2 following [4]. By integrating depth $d_i$ in Equation (1) along the ray, we can also obtain a expected depth value $\hat{D}(\mathbf{r})$:

$$\hat{D}(\mathbf{r}) = \sum_{i=1}^{N} d_i \alpha_i \prod_{j=1}^{i-1}(1 - \alpha_j).\tag{2}$$

## 4 Method

The overall framework is shown in Fig. 2. Given a video of a free-trajectory $\{I_i\}$ captured at time $\{t_i\}$ and the event stream $\varepsilon = \{\mathbf{e}_k\}$, our goal is to reconstruct the 3DGS of the scene and the corresponding camera trajectory $\{T_i\}$. Following the analysis-by-synthesis paradigm of 3DGS, we extend this approach by incorporating event camera data through two fundamental imaging principles: the Event Generation Model (**EGM**) and Linear Event Generation Model (**LEGM**). To address the absence of color information in events and ensure cross-view consistency, we further introduce photometric bundle adjustment (**PBA**) and a **Fixed-GS** training strategy.

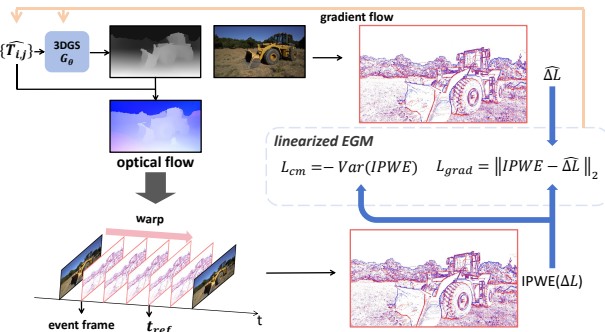

Figure 3: The illustration of unified CMax and LEGM optimization. We warp previous event frames to the sampled timestamp through the optical flow and maximize the sharpness of the image of IPWE. The byproduct IPWE is utilized to establish additional constraints on 3DGS.

## 4.1 EGM Driven Optimization

The EGM describes how event cameras asynchronously record pixel-wise brightness changes. When the logarithmic brightness change at a pixel $\mathbf{u}_k = (x_k, y_k)$, exceeds a predefined contrast threshold $C$,

$$\Delta L(\mathbf{u}_k, t_k) \doteq L(\mathbf{u}_k, t_k) - L(\mathbf{u}_k, t_k - \delta t) = p_k C, \tag{3}$$

where $L \doteq log(I)$ is the logarithm of intensity, $p_k \in \{-1, +1\}$ indicates the polarity of brightness changes, and $t_k$ is the triggered timestamp.

As shown in Fig. 2 (1.1), to leverage the high temporal resolution of events, we first divide the time interval between two adjacent video frames $I_i$ and $I_{i+1}$ into N smaller subintervals $\varepsilon_{i,j} = \{\mathbf{e}_k | t_{i,j} \leq t_k \leq t_{i,j+1}, \Delta t = \frac{t_{i+1} - t_i}{N}, t_{i,j} = t_i + j \cdot \Delta t\}$. This allows us to form accumulated event frames at a higher temporal resolution:

$$E_{i,j} = \sum_{e_k \in \varepsilon_{i,j}} p_k. \tag{4}$$

We then reconstruct the latent intensity image $I_t$ at any intermediate time $t \in \{t_{i,j}\}$ by integrating the accumulated events with the most recent frame:

$$I_t = I_{i,j} = \begin{cases} I_{i,0} \cdot \exp(\sum_{n=0}^{j-1} E_{i,n} \cdot C) & \text{if } j > 0 \\ I_{i,0} & \text{if } j = 0 \end{cases}. \tag{5}$$

This latent intensity image provides a supervisory signal for our event-based rendering loss:

$$\mathcal{L}_{EGM} = (1 - \lambda)\mathcal{L}_1(\hat{I}_t, I_t) + \lambda \mathcal{L}_{D-SSIM}(\hat{I}_t, I_t). \tag{6}$$

By enforcing consistency between rendered and latent intensity images, this loss effectively utilizes the brightness information encoded in event streams between adjacent frames, addressing the challenge of sparse viewpoints in high-speed scenarios.

## 4.2 Unified CMax and LEGM Optimization

While $\mathcal{L}_{EGM}$ leverages the brightness change information recorded by events, it does not explicitly exploit the motion information encoded in the event stream. To address this, we introduce the Contrast Maximization (CMax) [27, 46, 47, 48] framework and the LEGM [28, 29, 49]. These models complement the previous EGM-driven optimization.

Under constant scene illumination, events are triggered by the motion of scene edges, forming continuous trajectories in (x, y, t) space. As shown in Fig. 2 (1.2), by warping(back-projecting) events along the correct motion trajectories, we can obtain a sharp image of warped events (IWE). Therefore, the sharpness of the IWE can serve as an indication of the accuracy of the estimated motion. This insight motivates us to derive the motion field by leveraging the rendered depth from 3DGS using eq. (2) and the relative camera motion between neighboring timestamps. By optimizing the sharpness of the IWE, we can obtain the optimal motion field, which in turn helps to improve the geometric accuracy of the 3DGS and the camera poses.

As shown in Fig. 3, for efficiency, we adopt a piece-wise warping approach instead of warping individual events. Specifically, for current timestamp $t_{ref} = t_{i,j}$, we warp the event frames from previous $r$ sub-intervals:

$$E_{i,j-m \to j} = \text{warp}(E_{i,j-m}, F_{i,j \to j-m}), \tag{7}$$

where $m \in [0, r]$, $F_{i,j-m \to j}$ is the optical flow derived from the rendered depth $\hat{D}$ in eq. (2) and relative pose $T_{i,j \to j-m}$ between two timestamps:

$$F_{i,j \to j-m} = \Pi(T_{i,j \to j-m} \Pi^{-1}(x, y, \hat{D})) - (x, y), \tag{8}$$

where $T_{i,j \to j-m} = T_{i,j-m} T_{i,j}^{-1}$, $\Pi$ projects a 3D point to image coordinates and $\Pi^{-1}$ unprojects a pixel coordinate and depth into a 3D point. Then the image of piece-wise warped events (IPWE) at timestamp $t_{i,j}$ is computed by averaging the warped event frames:

$$\text{IPWE}_{i,j} = \frac{1}{r+1} \sum_{m=j-r}^{j} E_{i,m \to j} \approx \frac{1}{C} \Delta L. \tag{9}$$

Following the Cmax framework, we maximize the variance of the IPWE, which is equivalent to minimize its opposite:

$$\mathcal{L}_{cm} = -\text{Var}(\text{IPWE}_{i,j}). \tag{10}$$

Furthermore, based on the LEGM [28, 29], the brightness change $\Delta L$ at pixel $\mathbf{u}$ can be approximated by the dot product of the image gradient $\nabla L$ and the optical flow $\dot{\mathbf{u}}$ (note that $L$ is the logarithm of an image):

$$\Delta L(\mathbf{u}) = -\nabla L \cdot \dot{\mathbf{u}} \approx L(\mathbf{u}) - L(\mathbf{u} + \dot{\mathbf{u}}). \tag{11}$$

It is noteworthy that the IPWE also encodes brightness change information. Combining eq. (9) and eq. (11), we establish a connection between the IPWE and the brightness changes of the rendered images:

$$C \cdot \text{IPWE}_{i,j} = \hat{L}(\mathbf{u}) - \hat{L}(u + F_{i,j \to j+1}). \tag{12}$$

Note that to compute $F_{i,j \to j+1}$, we estimate $T_{i,j+1}$ by leveraging the assumption of locally linear motion from $T_{i,j-1}$ and $T_{i,j}$. Based on this relationship, we formulate an additional gradient-based loss:

$$\mathcal{L}_{grad} = ||C \cdot \text{IPWE}_{i,j} - (\hat{L}(\mathbf{u}) - \hat{L}(u + F_{i,j \to j+1}))||^2, \tag{13}$$

where $\hat{L}$ is the logarithm of synthesised image $\hat{I}_t$. Finally, the full LEGM loss is defined as:

$$\mathcal{L}_{LEGM} = \lambda_{cm} \mathcal{L}_{cm} + \lambda_{grad} \mathcal{L}_{grad}, \tag{14}$$

where $\lambda_{cm}$ and $\lambda_{grad}$ are the balancing weight.

### 4.3 Photometric Bundle Adjustment

The aforementioned event-based constraints, $\mathcal{L}_{EGM}$ and $\mathcal{L}_{LEGM}$, leverage the brightness change and motion information encoded in the event data to constrain 3DGS. However, as event cameras only record brightness changes and lack color perception, directly applying them to 3DGS optimization may lead to inconsistent color rendering. To ensure cross-view consistency of the 3DGS rendering, we introduce the Photometric Bundle Adjustment (PBA) term.

Specifically, as shown in Fig. 2 (1.3), for a randomly sampled timestamp $t \in \{t_{i,j}\}$, we establish the following photometric reprojection error:

$$\mathcal{L}_{PBA} = \sum_{\mathbf{u} \in \mathcal{P}} \sum_{I_s \in \mathcal{F}} ||I_s(\mathbf{u}') - \hat{I}(\mathbf{u})||^2, \tag{15}$$

where $\mathbf{u}' = \Pi(T_{i,j-r \to j} \Pi^{-1}(x, y, \hat{D}(u)))$ represent the coordinate on target view projected from the pixel $\mathbf{u}$ of source view $I_s$, $\mathcal{P}$ denotes the pixel samples of current frame, and $\mathcal{F}$ is the candidates of target video frames. We select $\mathcal{F}$ to be the nearest previous video frame in consideration of computation costs.

| Methods | Pose-Free | Input | 6 FPS | | | 4 FPS | | | 3 FPS | | | 2 FPS | | | 1 FPS | | |
|---|---|---|---|---|---|---|---|---|---|---|---|---|---|---|---|---|---|
| | | | PSNR↑ | SSIM↑ | LPIPS↓ | PSNR↑ | SSIM↑ | LPIPS↓ | PSNR↑ | SSIM↑ | LPIPS↓ | PSNR↑ | SSIM↑ | LPIPS↓ | PSNR↑ | SSIM↑ | LPIPS↓ |
| F2-NeRF | × | F | 23.55 | 0.75 | 0.34 | 22.97 | 0.72 | 0.36 | 22.25 | 0.69 | 0.40 | 21.64 | 0.68 | 0.44 | 20.63 | 0.64 | 0.51 |
| Nope-NeRF | ✓ | F | 13.86 | 0.51 | 0.67 | 13.81 | 0.51 | 0.67 | 13.79 | 0.51 | 0.67 | 13.50 | 0.51 | 0.68 | 13.72 | 0.51 | 0.68 |
| LocalRF | ✓ | F | 23.94 | 0.73 | 0.36 | 23.05 | 0.71 | 0.39 | 22.49 | 0.69 | 0.40 | 21.20 | 0.66 | 0.44 | 19.42 | 0.63 | 0.48 |
| CF-3DGS | ✓ | F | 26.05 | 0.78 | 0.31 | 25.03 | 0.77 | 0.33 | 23.73 | 0.74 | 0.36 | 22.08 | 0.68 | 0.42 | 20.53 | 0.65 | 0.46 |
| EvDeblurNeRF | × | E+F | 22.43 | 0.71 | 0.38 | 21.23 | 0.69 | 0.42 | 20.09 | 0.65 | 0.49 | 17.52 | 0.62 | 0.55 | 15.19 | 0.55 | 0.60 |
| ENeRF | × | E+F | 23.62 | 0.73 | 0.37 | 22.84 | 0.70 | 0.38 | 21.85 | 0.69 | 0.41 | 20.52 | 0.66 | 0.46 | 18.09 | 0.60 | 0.52 |
| Event-3DGS(E+F) | × | E+F | 26.32 | 0.78 | 0.33 | 25.37 | 0.76 | 0.34 | 24.59 | 0.75 | 0.37 | 23.44 | 0.72 | 0.38 | 22.41 | 0.69 | 0.39 |
| EvCF-3DGS | ✓ | E+F | 26.07 | 0.78 | 0.32 | 25.48 | 0.77 | 0.33 | 24.61 | 0.75 | 0.36 | 22.81 | 0.70 | 0.38 | 21.73 | 0.67 | 0.43 |
| EF-3DGS(Ours) | ✓ | E+F | **26.66** | **0.79** | **0.30** | **26.01** | **0.78** | **0.30** | **25.38** | **0.77** | **0.31** | **24.43** | **0.74** | **0.34** | **23.96** | **0.72** | **0.36** |

Table 1: Quantitative evaluations on Tanks and Temples dataset. The best results are highlighted in bold. Note that, in the "input" column, "F" denotes traditional frame input, while "E+F" denotes hybrid frame and event input.

By minimizing $\mathcal{L}_{PBA}$ across sampled views, we encourage the 3DGS model to produce geometrically and photometrically consistent renderings across events and video frames, thus effectively resolving color inconsistencies inherent in event data.

## 4.4 Fixed-GS Training Strategy

The $\mathcal{L}_{PBA}$ term alone is insufficient to fully mitigate color distortion issues. To further address this challenge, we propose a two-stage Fixed-GS scene optimization strategy that takes advantage of 3DGS's explicit attribute representation. In the first stage, all the parameters are optimizable and the optimization is performed across all timestamps:

$$G_\theta^*, T_{i,j}^* = \underset{\mu,\alpha,r,s,f,T_{i,j}}{\operatorname{argmin}} \mathcal{L}_{event}, \ t \in \{t_{i,j}\}, \tag{16}$$

where $\mu, \alpha, r, s, f$ is the position, opacity, rotation, scale factor and spherical harmonics of the Gaussians, and t is the sampled timestamp during training. This stage results in a scene reconstruction with accurate structure and brightness, albeit with potential color distortions due to the dominant colorless event supervision overwhelming the sparse RGB frame color supervision. The second stage focuses on recovering accurate color information. During this phase, optimization is conducted exclusively on video frames. We optimize only the spherical harmonic coefficients of the Gaussians while keeping other parameters fixed:

$$G_\theta^* = \underset{f}{\operatorname{argmin}} \mathcal{L}_{color}, \ t \in \{t_{i,0}\} \tag{17}$$

The ratio between the first and second stages is empirically set to 4:1. This approach allows us to effectively address the color distortion problem while preserving the structural and brightness information obtained from the event data.

## 4.5 Overall Training Pipeline

Assembling all loss terms, we get the overall loss function:

$$\mathcal{L}_{event} = \mathcal{L}_{EGM} + \mathcal{L}_{LEGM} + \lambda_{PBA}\mathcal{L}_{PBA}, \tag{18}$$

where $\lambda_{PBA}$ are the weighting factor. Note that since event cameras typically record only the changes in brightness intensity, the $\mathcal{L}_{EGM}$ and $\mathcal{L}_{LEGM}$ losses are computed in the grayscale domain, whereas the $\mathcal{L}_{PBA}$ loss is calculated in RGB color space. We incorporate dynamic scene allocation strategies from LocalRF [8] for handling extended video sequences. Our overall training pipeline builds upon the progressive optimization scheme of CF-3DGS [5] while introducing novel components to integrate event stream data for robust free-trajectory scene reconstruction. Please refer to Section A.3 for the algorithm pipeline and additional implementation details.

# 5 Experiments

## 5.1 Dataset

**Tanks and Temples.** We conduct comprehensive experiments on the Tanks and Temples dataset [50]. Similar to LocalRF [8], we adopt 9 scenes, covering large-scale indoor and outdoor scenes. For each scene, we sample a video clip with a 50-second duration, typically featuring free camera trajectories

| Methods | Input | 6 FPS | | | 4 FPS | | | 3 FPS | | | 2 FPS | | | 1 FPS | | |
|---|---|---|---|---|---|---|---|---|---|---|---|---|---|---|---|---|
| | | RPE$_t\downarrow$ | RPE$_r\downarrow$ | ATE$\downarrow$ | RPE$_t\downarrow$ | RPE$_r\downarrow$ | ATE$\downarrow$ | RPE$_t\downarrow$ | RPE$_r\downarrow$ | ATE$\downarrow$ | RPE$_t\downarrow$ | RPE$_r\downarrow$ | ATE$\downarrow$ | RPE$_t\downarrow$ | RPE$_r\downarrow$ | ATE$\downarrow$ |
| Nope-NeRF | F | 0.1141 | 0.7563 | 2.8382 | 0.1604 | 1.0542 | 2.7653 | 0.2220 | 1.3694 | 2.7857 | 0.3131 | 1.8965 | 2.8412 | 0.6216 | 3.913 | 2.6592 |
| LocalRF | F | 0.0806 | 0.9282 | 0.5630 | 0.0867 | 0.9683 | 0.6085 | 0.0911 | 0.9800 | 0.6501 | 0.0957 | 1.0428 | 0.6802 | 0.1421 | 1.4725 | 1.0006 |
| CF-3DGS | F | 0.0594 | 0.6981 | 0.4212 | 0.0637 | 0.7128 | 0.4628 | 0.0712 | 0.7531 | 0.5189 | 0.0859 | 0.8074 | 0.6918 | 0.1057 | 0.9768 | 0.8972 |
| EvCF-3DGS | E+F | 0.0461 | 0.5972 | 0.3419 | 0.0490 | 0.6269 | 0.3766 | 0.0538 | 0.6728 | 0.4261 | 0.0591 | 0.7094 | 0.4860 | 0.0657 | 0.7597 | 0.5534 |
| EF-3DGS(Ours) | E+F | **0.0391** | **0.5427** | **0.2885** | **0.0407** | **0.5521** | **0.3064** | **0.0426** | **0.5796** | **0.3271** | **0.0449** | **0.5953** | **0.3671** | **0.0487** | **0.6259** | **0.3753** |

Table 2: Pose accuracy on Tanks and Temples. We use COLMAP poses in Tanks and Temples as the "ground truth". The unit of RPE$_r$ is in degrees, ATE is in the ground truth scale and RPE$_t$ is scaled by 100. Those methods that require precomputed poses are excluded.

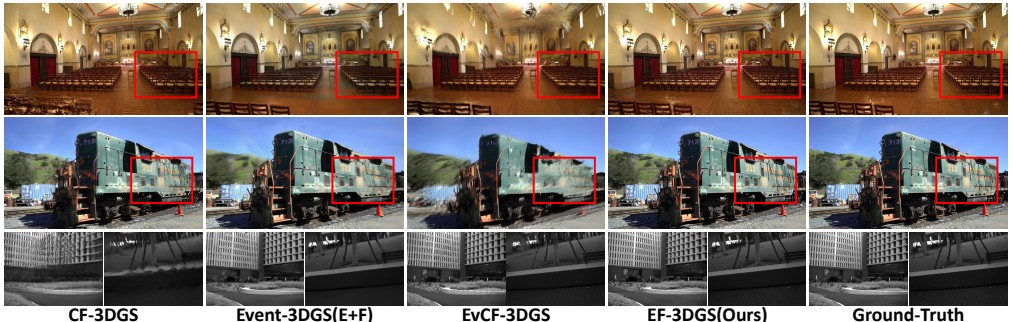

| CF-3DGS | Event-3DGS(E+F) | EvCF-3DGS | EF-3DGS(Ours) | Ground-Truth |

Figure 4: Qualitative comparison for novel view synthesis. The first two rows are from Tanks and Temples and the last row is from RealEv-DAVIS. Our approach produces more realistic rendering results with fine-grained details. Better viewed when zoomed in.

and covering a considerable distance. Following LocalRF [8], we apply 4× spatial downsampling to the videos. To evaluate the robustness under varying camera speeds, we employ varying temporal downsampling of 6 FPS, 4 FPS, 3 FPS, 2 FPS, and 1 FPS. The reduction in frame rate effectively creates larger inter-frame displacements, simulating high-speed scenarios. To synthesize realistic event data, we first upsample the original videos by [51] and then apply the simulator V2E [52].

**RealEv-DAVIS.** Due to the lack of free-trajectory event camera datasets, we introduce RealEv-DAVIS. Using a DAVIS346 camera that simultaneously captures frames and events at 346×260 resolution, we record 40-second handheld sequences at 25 FPS. We employ COLMAP for ground-truth poses. For SLOW scenarios, we retain every second frame, while for FAST scenarios, we keep only one frame per five frames. Further details are provided in Section A.2.

## 5.2 Implementation details

We follow the optimization parameters by the configuration outlined in the 3DGS [4]. We optimize the camera poses in the representation of quaternion rotation. The initial learning rate is set to $10^{-5}$ and gradually decays to $10^{-6}$ until convergence. The balancing weight $\lambda_{cm}$, $\lambda_{grad}$ and $\lambda_{PBA}$ is empirically set to 0.1, 0.2 and 0.5. For the division of events between adjacent frames, we maintain a constant interval of $\frac{1}{6}$s for Tanks and Temples and $\frac{1}{25}$s for RealEv-DAVIS, setting the number of subinterval N accordingly. For example, in Tanks and Temples, N equals 2 for 3FPS and 6 for 1FPS. This ensures adherence to the constant brightness assumption within each sub-interval and provides adequate events for the following CMax warping. The intervals of neighboring warping $r$ in CMax are set to 3. The contrast threshold $C$ is set to 0.25 for Tanks and Temples and 0.21 for RealEv-DAVIS. We provide detailed ablation studies on these hyperparameters and additional implementation details in Section A.4.

## 5.3 Experimental Setup

**Metrics.** We evaluate all the methods from two aspects: novel view synthesis and pose estimation. For the novel view synthesis task, we report the standard metrics PSNR, SSIM [53], and LPIPS [54]. For the pose estimation task, we adopt the Absolute Trajectory Error (ATE) and Relative Pose Error (RPE) metrics [55, 56], as delineated in [10]. Since these metrics are inherently influenced by frame rate, we upsample all estimated poses to a consistent temporal resolution before evaluation for fair comparison across different frame rate settings.

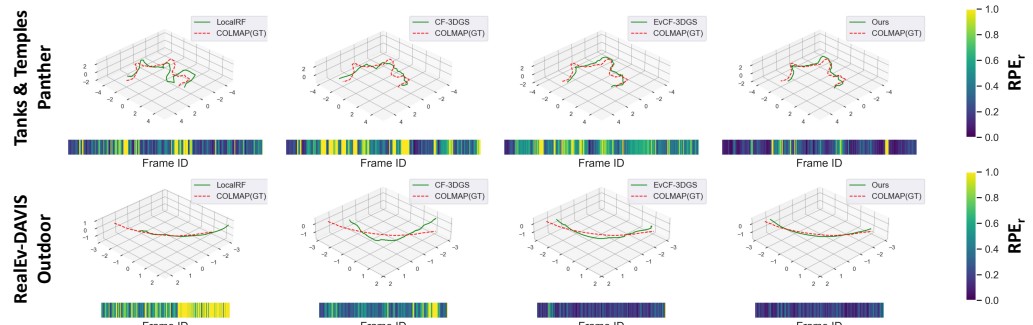

Figure 5: Pose estimation comparison. We visualise the trajectory (3D plot) and $\text{RPE}_r$ (color bar) of each method. We clip and normalize the $\text{RPE}_r$ by a quarter of the max $\text{RPE}_r$ across all results of each scene.

| Methods | Input | SLOW | | | | FAST | | | |
|---|---|---|---|---|---|---|---|---|---|
| | | NVS | | Pose | | NVS | | Pose | |
| | | PSNR↑ | SSIM↑ | RPE$_t$↓ | RPE$_r$↓ | PSNR↑ | SSIM↑ | RPE$_t$↓ | RPE$_r$↓ |
| LocalRF | F | 20.83 | 0.6074 | 3.60 | 2.07 | 17.62 | 0.5192 | 5.22 | 2.96 |
| CF-3DGS | F | 22.68 | 0.6287 | 2.49 | 1.55 | 17.59 | 0.5204 | 3.68 | 2.17 |
| EvDeblurNeRF | E+F | 20.61 | 0.6064 | - | - | 17.98 | 0.5269 | - | - |
| Event-3DGS (E+F) | E+F | 23.43 | 0.6456 | - | - | 20.04 | 0.5515 | - | - |
| EvCF-3DGS | E+F | 22.89 | 0.6317 | 1.78 | 0.82 | 19.13 | 0.5380 | 2.70 | 1.28 |
| EF-3DGS(Ours) | E+F | **23.65** | **0.6466** | **1.41** | **0.69** | **21.12** | **0.5620** | **1.80** | **0.89** |

Table 3: Rendering and pose estimation results on RealEv-DAVIS. Complete data and additional metrics are provided in the supplementary material.

**Baselines.** For a fair comparison, we focus on two categories of methods: (1) For frame-based approaches, we selected methods specifically addressing free-trajectory scenarios, such as LocalRF [8] and F2-NeRF [57]. We also include pose-free methods like Nope-NeRF [10] and CF-3DGS [5]. (2) For event-frame hybrid methods, we consider approaches that fuse events and frames, including ENeRF [36], EvDeblurNeRF [40] and Event-3DGS [44]. Since no existing method integrates events for free-trajectory scenarios, we implement EvCF-3DGS as a competitive baseline that leverages an event-based frame interpolation network (Time Lens [58]) to temporally upsample frames before feeding them into CF-3DGS.

## 5.4 Experimental Results

We select every ten frames as a test image for NVS evaluation following LocalRF [8]. Since the camera poses are unknown in our setting, we need to estimate the poses of test views. As in iNeRF [59], we freeze the 3DGS model, initialize the test poses with the poses of the nearest training frames, and optimize the test poses by minimizing the photometric error between rendered images and test views.

**Results on RealEv-DAVIS.** Table 3 validates our approach on the real-world RealEv-DAVIS dataset. EF-3DGS outperforms top-performing methods and handles real-world scenes effectively. In FAST scenarios, our method shows nearly 1dB PSNR improvement over the best baselines. This confirms our advantage in high-speed scenarios where frame-based methods struggle. Fig. 4 and Fig. 5 show our method preserves fine details and maintains accurate trajectories even during rapid motion, addressing key limitations of traditional approaches.

| $\mathcal{L}_{EGM}$ | $\mathcal{L}_{LEGM}$ | $\mathcal{L}_{PBA}$ | Fixed GS | NVS | | | Pose | | |
|---|---|---|---|---|---|---|---|---|---|
| | | | | PSNR↑ | SSIM↑ | LPIPS↓ | RPE$_t$↓ | RPE$_r$↓ | ATE↓ |
| | | | | 20.53 | 0.65 | 0.46 | 0.1057 | 0.9768 | 0.8972 |
| ✓ | | | | 22.16 | 0.68 | 0.42 | 0.0651 | 0.7529 | 0.5779 |
| | ✓ | | | 21.07 | 0.67 | 0.44 | 0.0830 | 0.8869 | 0.7231 |
| | | ✓ | | 20.96 | 0.65 | 0.46 | 0.0938 | 0.9875 | 0.9112 |
| ✓ | ✓ | | | 22.83 | 0.68 | 0.40 | 0.0523 | 0.6387 | 0.3981 |
| ✓ | ✓ | ✓ | | 23.46 | 0.70 | 0.37 | 0.0523 | 0.6387 | 0.3981 |
| ✓ | ✓ | ✓ | | 23.09 | 0.70 | 0.38 | **0.0487** | **0.6259** | **0.3753** |
| ✓ | ✓ | ✓ | ✓ | **23.96** | **0.72** | **0.36** | **0.0487** | **0.6259** | **0.3753** |

Table 4: Effect of each component in EF-3DGS. The best results are highlighted in bold.

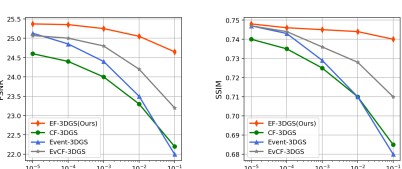

Figure 6: Robustness of different methods to pose disturbance.

**Results on Tanks and Temples.** Tables 1 and 2 demonstrate two key findings: (1) Our event-aided approach achieves up to 3dB higher PSNR and nearly 40% lower trajectory error at 1FPS compared to frame-based methods, indicating the critical value of event data in high-speed scenarios. (2) Our method maintains 1.55dB PSNR advantage over Event-3DGS at 1FPS, confirming that our integration framework effectively exploits the nature of event data beyond merely using it. Fig. 4 shows our method produces sharper edges and finer textures, while Fig. 5 illustrates we achieve more accurate trajectory estimation.

**Performance under Varying Camera Speeds** As shown in Table 2 and Table 3, while all methods degrade as the frame rate decreases, our approach shows remarkable resilience. The performance gap widens significantly at lower frame rates, with our PSNR advantage over CF-3DGS [5] increasing from 0.61dB at 6FPS to 3.43dB at 1FPS. Notably, our method also consistently outperforms other event-based methods (EvCF-3DGS and Event-3DGS). This confirms not only the value of event data in challenging scenarios but also the superiority of our integration approach.

## 5.5 Ablation Studies

**Effect of Each Component** Table 4 presents a comprehensive ablation study of our key components under the challenging 1FPS setting on Tanks and Temples. $\mathcal{L}_{EGM}$ serves as the foundation of our approach, providing substantial improvements in both rendering quality (+1.63dB PSNR) and pose accuracy by enabling rich supervision between discrete frames. Building upon this, $\mathcal{L}_{LEGM}$ extracts motion information from events and constrains 3DGS in the gradient domain, significantly improving pose estimation while modestly enhancing rendering quality. $\mathcal{L}_{PBA}$, though designed to address color inconsistency issues, not only improves rendering quality but also enhances pose estimation accuracy by establishing geometric and photometric consistency across views. The Fixed-GS training strategy, while having no impact on pose optimization, significantly improves rendering quality by effectively separating structure and color optimization. We provide more intuitive ablation visualizations in Section A.5.

**Robustness to Pose Disturbance** To validate the robustness of different methods under inaccurate pose initialization, a common challenge in practical scenarios, we introduce varying degrees of perturbations to the initial camera poses estimated by COLMAP. Specifically, following BARF [32], we parametrize the camera poses $\mathbf{p}$ with the $\mathfrak{se}(3)$ Lie algebra. For each scene, we synthetically perturb the camera poses with additive noise $\delta\mathbf{p} \sim \mathcal{N}(\mathbf{0}, n\mathbf{I})$, where n is the noise level. Then, each method is initialized with the noised poses, after which the optimization is performed. The results are illustrated in Fig. 6. Notably, Event-3DGS [44], which lacks the capability to optimize camera poses, exhibits a drastic performance degradation as the magnitude of pose disturbances increases. This observation validates the critical importance of joint pose-scene optimization. Furthermore, Our proposed framework demonstrates superior tolerance across all perturbation levels. Even under significant noise, our method experiences substantially less degradation in both rendering quality and trajectory accuracy.

## 6 Conclusions

In this work, we propose Event-Aided Free-Trajectory 3DGS (EF-3DGS), a novel framework that seamlessly integrates event camera data into the task of reconstructing 3DGS from casually captured free-trajectory videos. Our method effectively leverages the high temporal resolution and motion information encoded in event streams to enhance the 3DGS optimization process, leading to improved rendering quality and accurate camera pose estimation. By introducing the Event Generation Model and Linear Event Generation Model, we bridge the gap between differential event data and absolute brightness rendering in 3DGS. The proposed photometric bundle adjustment and Fixed-GS strategy further ensure accurate color recovery and scene structure optimization. Extensive experiments on both public benchmarks and real-world datasets validate the effectiveness of our approach, demonstrating significant improvements over state-of-the-art methods in both rendering quality and trajectory estimation accuracy. Future work would explore self-adaptive parameter adjustment strategies to enhance the method's versatility and ease of use across various reconstruction tasks.

## 7 Acknowledgements

This work is supported by the National Natural Science Foundation of China (NSFC) under Grants 62225207, 62436008 and 62306295. The AI-driven experiments, simulations and model training were performed on the robotic AI-Scientist platform of Chinese Academy of Sciences.

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

# A  Appendix

## A.1  Discussion on Motion Blur

Image blur arises from brightness integration during exposure time and occasionally occurs in low-light conditions or high-speed motion scenarios. While our method does not explicitly target image blur, this does not diminish its core contributions. We can draw an analogy to 2D vision tasks: image deblurring addresses blur-induced degradation, while video frame interpolation tackles discontinuity from excessive inter-frame motion. These represent distinct challenges in computer vision. Similarly, in 3D reconstruction, **blur-related degradation and the challenges of sparse viewpoints with pose estimation constitute separate problem domains that often co-occur in high-speed scenarios but require distinct technical solutions.** Our work specifically addresses the latter—fundamental issues that persist regardless of blur conditions and represent critical bottlenecks in high-speed 3D reconstruction.

**Previous event-based scene reconstruction methods have primarily focused on blur reconstruction while overlooking the challenges of sparse viewpoints and inaccurate pose estimation.** Our approach targets this gap by leveraging a novel fusion strategy to enhance traditional image-based 3DGS reconstruction, addressing fundamental limitations that affect reconstruction quality independent of blur artifacts.

For completeness, we conducted supplementary experiments to evaluate our method's performance under motion blur conditions. We extend our method to handle motion blur by incorporating the Event Double Integration (EDI) model [41], which reconstructs sharp intensity from event data. Specifically, we reformulate the intensity term $I_{i,0}$ in Eq. (5) using the EDI formulation:

$$\hat{I}_{i,0} = \frac{(2n+1) \cdot B_i}{\sum_{k=-n}^{n} \exp\left(C \cdot \sum_{z=0}^{k} E_{i,z}\right)}, \tag{19}$$

where $B_i$ represents the blurred intensity, $E_{i,z}$ denotes the accumulated events, $C$ is the contrast threshold, and $n$ defines the temporal integration window of blur averaging.

**Experimental Setup:** Experiments were performed on multiple scenes from the Tanks and Temples [50] with a frame rate of 2FPS. Motion-blurred frames were synthesized adopting the blur generation protocol from the GoPro-Blur dataset through gamma correction and multi-frame averaging operations. We average every 30 frames to simulate blurring. We selected EvDeblurNeRF [40], currently the best-performing open-source event-based deblur scene reconstruction method, as our comparison.

**Results:** As demonstrated in Tab. 5, our method achieves comparable reconstruction quality to EvDeblurNeRF while simultaneously performing pose estimation. Visual comparisons in Fig. 7 show that our method renders sharper novel views under motion blur conditions. Note that EvDeblurNeRF requires pre-computed COLMAP poses. This limitation significantly restricts practical applicability in high-speed scenarios where accurate pose estimation is challenging. These results demonstrate that our method maintains robust performance under blur conditions while addressing the fundamental pose estimation and sparse-view challenge.

Table 5: Comparison of different methods on pose estimation and rendering quality.

| Methods | Pose Estimation | PSNR↑ | SSIM↑ | $RPE_t\downarrow$ | $RPE_r\downarrow$ |
|---|---|---|---|---|---|
| EvDeblurNeRF | CF-3DGS | 21.17 | 0.69 | 0.105 | 0.897 |
| EvDeblurNeRF | COLMAP | 22.58 | 0.70 | - | - |
| EDI + EF-3DGS (Ours) | EDI + EF-3DGS (Ours) | **23.18** | **0.72** | **0.062** | **0.642** |

## A.2  Dataset

For the synthetic dataset, following LocalRF [8], we choose nine static scenes from the Tanks and Temples dataset, which cover both indoor and outdoor environments. To construct a real-world dataset, we utilized a handheld DAVIS346 event camera to capture a series of extended video sequences,

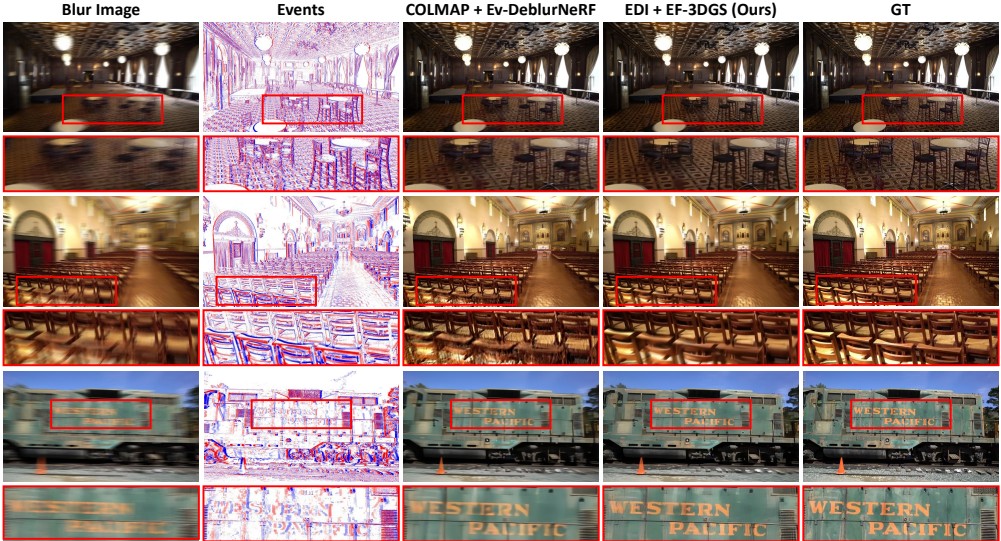

| Blur Image | Events | COLMAP + Ev-DeblurNeRF | EDI + EF-3DGS (Ours) | GT |

Figure 7: Qualitative comparison on motion deblur scene reconstruction. Our EDI + EF-3DGS method produces sharper results comparable to EvDeblurNeRF

simulating free camera trajectories. Particularly for the scenes named building, we deliberately introduced significant camera motion during acquisition to emulate realistic camera shake scenarios. Details about these sequences are illustrated in Table 6, where Mean. rotation represents the mean relative rotation angle between two adjacent frames and Max. rotation denotes the maximum relative rotation angle between two frames in a sequence. We select a 50-second segment from each sequence to highlight scenarios with free-trajectory camera movements.

Table 6: Details of selected sequences on Tanks and Temples [50] and RealEv-DAVIS.

| | Scenes | Seq. Length | Mean. rotation (deg) | Max. rotation (deg) |
|---|---|---|---|---|
| Tanks and Temples | Auditorium | 300 | 2.91 | 54.55 |
| | Ballroom | 300 | 5.73 | 179.81 |
| | Caterpillar | 300 | 3.18 | 102.37 |
| | Church | 300 | 2.44 | 37.22 |
| | Courtroom | 300 | 8.40 | 177.96 |
| | M60 | 300 | 5.29 | 179.99 |
| | Museum | 300 | 6.05 | 176.98 |
| | Panther | 300 | 4.33 | 124.63 |
| | Train | 300 | 4.80 | 108.18 |
| RealEv-DAVIS | building | 500 | 5.65 | 67.65 |
| | hall | 500 | 3.88 | 106.73 |
| | corner | 500 | 3.04 | 96.11 |
| | outdoor | 500 | 2.36 | 77.18 |

### A.3 Overall Training Pipeline

We detail the comprehensive training pipeline in Algorithm 1. Note that for notational simplicity, we consolidate the subscripts $(i, j)$ into a single index $k$. Our approach incorporates the dynamic radiance field allocation strategy from LocalRF [8], which assigns a new radiance field when the current pose exceeds a distance threshold of 1 from the existing field. However, in practical implementations, we encountered the OOM(out of memory) problem with this approach. To address this, we adopted a modified strategy of allocating a new 3DGS every 50 frames, based on the sequence length. This adjustment ensures efficient memory utilization while maintaining the benefits of dynamic allocation. For pose optimization and scene reconstruction, we adopt strategies from CF-3DGS.

And we incorporate event-driven brightness and motion coherence constraints, enabling robust reconstruction in challenging high-speed scenarios.

---

**Algorithm 1** Overall Training Pipeline.

---

1: **Input**: event frames $\{I_k\}_{k=0}^{K} (k = i \cdot N + j)$.
2: **Output**: camera poses $\{T_k\}_{k=1}^{K}$, 3DGS $\{G_{\theta_n}\}$.
3: Initialize current center frame of 3DGS, $l = 0$;
4: **while** $l < K$ **do**                                                       ▷ Not finish the optimization yet
5:     $G_{\theta_n} \leftarrow$ Initialize($I_l$) ▷ Following [5], initialize **global** 3DGS using the current center frame $I_l$
6:     **for** k in $[l + 1, min(l + 50, K)]$ **do**   ▷ Optimize the 3DGS and the poses of next 50 frames
7:
8:         *# Pose Estimation Step*
9:             $G_{local} \leftarrow$ Initialize($I_{k-1}$)        ▷ Following [5], initialize **local** 3DGS using the previous frame $I_{k-1}$
10:            $T_k \leftarrow$ Optimize($\mathcal{L}_{event}, T_k$)        ▷ estimate camera pose $T_k$ by minimizing $\mathcal{L}_{event}$ at $T_k$
11:    **end for**
12:
13:        *# Scene Reconstruction Step*
14:        $\{G_{\theta_n}\} \leftarrow$ Optimize($\mathcal{L}_{event}, T_{[l,l+50]}$)        ▷ Optimizing 3DGS $G_{\theta_n}$ by minimizing $\mathcal{L}_{event}$ at $T_{[l,l+50]}$
15:        SHs of $\{G_{\theta_n}\} \leftarrow$ Optimize($\mathcal{L}_{color}, T_{[l,l+50]}$)   ▷ Fixed-GS stage: optimizing SHs of 3DGS $G_{\theta_n}$ by minimizing $\mathcal{L}_{color}$ at RGB frames
16:        $l \leftarrow k$                               ▷ Slide the current center frame to the next center frame
17: **end while**

---

Table 7: Ablation study results of parameter $r$ in CMax framework

| $r$ | NVS | | | Pose | | |
|---|---|---|---|---|---|---|
| | PSNR ↑ | SSIM ↑ | LPIPS ↓ | RPE$_t$ ↓ | RPE$_r$ ↓ | ATE ↓ |
| 1 | 22.94 | 0.70 | 0.39 | 0.0591 | 0.6829 | 0.4851 |
| 3 | **23.96** | **0.72** | **0.36** | **0.0487** | **0.6259** | **0.3753** |
| 5 | 23.49 | **0.72** | 0.37 | 0.0524 | 0.6431 | 0.4338 |

## A.4 Additional Experiments

### A.4.1 Detailed Experiment Results on RealEv-DAVIS

Due to space constraints in the main text, we provide a detailed table of our method's performance on the RealEv-DAVIS dataset in the supplementary materials, as shown in Tab. 8 and Tab. 9.

Table 8: Quantitative Evaluations on RealEv-DAVIS. We select the top-performing methods from the previous evaluation. Each baseline method is trained with its public code under the original settings and evaluated with the same evaluation protocol. The best results are highlighted in bold.

| Methods | Pose-Free | Input | SLOW | | | | | | | | | | | | FAST | | | | | | | | | | | |
|---|---|---|---|---|---|---|---|---|---|---|---|---|---|---|---|---|---|---|---|---|---|---|---|---|---|---|
| | | | hall | | | building | | | corner | | | outdoor | | | hall | | | building | | | corner | | | outdoor | | |
| | | | PSNR↑ | SSIM↑ | LPIPS↓ | PSNR↑ | SSIM↑ | LPIPS↓ | PSNR↑ | SSIM↑ | LPIPS↓ | PSNR↑ | SSIM↑ | LPIPS↓ | PSNR↑ | SSIM↑ | LPIPS↓ | PSNR↑ | SSIM↑ | LPIPS↓ | PSNR↑ | SSIM↑ | LPIPS↓ | PSNR↑ | SSIM↑ | LPIPS↓ |
| LocalRF | ✓ | F | 18.76 | 0.5589 | 0.65 | 20.33 | 0.5751 | 0.45 | 21.09 | 0.5809 | 0.43 | 23.13 | 0.7146 | 0.36 | 17.89 | 0.4935 | 0.65 | 16.33 | 0.4607 | 0.63 | 17.27 | 0.4859 | 0.58 | 19.01 | 0.6369 | 0.46 |
| CF-3DGS | ✓ | F | 22.37 | 0.5990 | 0.51 | 21.33 | 0.5848 | 0.39 | 22.48 | 0.5976 | 0.42 | 24.52 | 0.7334 | 0.29 | 17.33 | 0.4923 | 0.66 | 16.92 | 0.4650 | 0.66 | 16.67 | 0.4835 | 0.59 | 19.42 | 0.6407 | 0.43 |
| EvDeblurNeRF | × | E+F | 20.24 | 0.5762 | 0.55 | 19.35 | 0.5649 | 0.50 | 20.47 | 0.5757 | 0.44 | 22.36 | 0.7087 | 0.38 | 18.23 | 0.5043 | 0.65 | 16.41 | 0.4624 | 0.63 | 17.82 | 0.4959 | 0.59 | 19.44 | 0.6448 | 0.42 |
| ENeRF | × | E+F | 21.89 | 0.5933 | 0.52 | 21.94 | 0.5918 | 0.36 | 22.01 | 0.5918 | 0.42 | 23.91 | 0.7266 | 0.30 | 20.09 | 0.5261 | 0.62 | 18.99 | 0.4903 | 0.50 | 19.61 | 0.5173 | 0.52 | 20.00 | 0.6494 | 0.42 |
| Event-3DGS | × | E+F | 23.01 | 0.6085 | 0.47 | 22.45 | 0.6191 | 0.34 | 23.11 | 0.6148 | 0.41 | 25.14 | 0.7399 | 0.23 | 20.35 | 0.5328 | 0.62 | 18.52 | 0.4907 | 0.51 | 20.05 | 0.5231 | 0.51 | 21.24 | 0.6596 | 0.40 |
| EvCF-3DGS | ✓ | E+F | 22.58 | 0.6014 | 0.48 | 21.71 | 0.5878 | 0.38 | 22.61 | 0.6013 | 0.42 | 24.66 | 0.7363 | 0.25 | 19.01 | 0.5139 | 0.64 | 17.99 | 0.4766 | 0.59 | 18.81 | 0.5061 | 0.55 | 20.70 | 0.6554 | 0.44 |
| EF-3DGS(Ours) | ✓ | E+F | 23.43 | 0.6103 | 0.47 | 22.30 | 0.6094 | 0.35 | 23.38 | 0.6210 | 0.41 | 25.50 | 0.7458 | 0.23 | 21.14 | 0.5386 | 0.60 | 20.05 | 0.5016 | 0.47 | 20.68 | 0.5294 | 0.50 | 22.61 | 0.6783 | 0.39 |

### A.4.2 Evaluating the Influence of parameter r in LEGM

We investigate the impact of the parameter $r$ in the Contrast Maximization (CMax) framework, which determines the number of previous event frames warped to the current sampled timestamp. Table 7 presents the results of this ablation study. The optimal value for $r$ is found to be 3, yielding the best

Table 9: Pose accuracy on RealEv-DAVIS. The unit of $RPE_r$ is in degrees, ATE is in the ground truth scale and $RPE_t$ is scaled by 100. Those methods which require precomputed poses are excluded.

| Methods | Input | SLOW | | | | | | | | | | | | FAST | | | | | | | | | | | |
|---|---|---|---|---|---|---|---|---|---|---|---|---|---|---|---|---|---|---|---|---|---|---|---|---|---|
| | | hall | | | building | | | corner | | | outdoor | | | hall | | | building | | | corner | | | outdoor | | |
| | | $RPE_t\downarrow$ | $RPE_r\downarrow$ | ATE↓ | $RPE_t\downarrow$ | $RPE_r\downarrow$ | ATE↓ | $RPE_t\downarrow$ | $RPE_r\downarrow$ | ATE↓ | $RPE_t\downarrow$ | $RPE_r\downarrow$ | ATE↓ | $RPE_t\downarrow$ | $RPE_r\downarrow$ | ATE↓ | $RPE_t\downarrow$ | $RPE_r\downarrow$ | ATE↓ | $RPE_t\downarrow$ | $RPE_r\downarrow$ | ATE↓ | $RPE_t\downarrow$ | $RPE_r\downarrow$ | ATE↓ |
| LocalRF | F | 4.4017 | 2.5067 | 0.2705 | 4.6186 | 1.9409 | 0.4909 | 1.9296 | 2.5183 | 0.4177 | 3.4307 | 1.3005 | 0.1984 | 5.6061 | 3.2265 | 0.3250 | 7.0611 | 2.8722 | 0.6853 | 3.4767 | 3.8950 | 0.5693 | 4.7229 | 1.8595 | 0.2798 |
| CF-3DGS | F | 2.7106 | 0.6035 | 0.1459 | 3.0374 | 3.2761 | 0.2142 | 1.9437 | 1.4954 | 0.2225 | 2.2747 | 0.8053 | 0.0865 | 3.5025 | 0.7639 | 0.2635 | 4.6758 | 4.5256 | 0.4640 | 3.4950 | 2.2855 | 0.3482 | 3.0551 | 1.1198 | 0.1909 |
| Ev-Baseline | E+F | 0.7063 | 0.4386 | 0.1218 | 3.4225 | 1.1717 | 0.1988 | 2.1283 | **1.3361** | **0.1565** | 0.8647 | **0.3455** | 0.0857 | 0.8898 | 0.7734 | 0.1307 | 5.0707 | 1.6997 | 0.3166 | 3.6785 | **2.1552** | 0.2121 | 1.1645 | 0.5001 | 0.1556 |
| EF-3DGS(Ours) | E+F | **0.5262** | **0.3251** | **0.1041** | **2.8008** | **0.4633** | **0.1737** | **1.8386** | 1.6168 | 0.1864 | **0.4726** | 0.3699 | **0.0749** | **0.5424** | **0.3464** | 0.1369 | **3.5848** | **0.5589** | **0.2645** | **2.5396** | 2.2157 | **0.2029** | **0.5218** | 0.4462 | **0.1136** |

performance across all metrics for both novel view synthesis and pose estimation. When $r = 1$, the performance degrades significantly, likely due to insufficient temporal information for accurate motion estimation. Increasing $r$ to 5 leads to a performance decline, though less severe than $r = 1$. This degradation at $r = 5$ can be attributed to the violation of the local linear motion assumption, which is fundamental to the CMax framework.

### A.4.3  Computational Efficiency and Impact of Subinterval Number N

To investigate the impact of the number of subintervals N on our method, we conduct ablation studies on the Tanks and Temples dataset under the 2 FPS setting. The results are shown in Table 10 and Fig. 8. We test the speed on RTX2080ti. Note that in the t-PSNR figure in Fig 8, circular nodes represent pose-free methods, while diamond-shaped nodes indicate methods that require precomputed poses. The results in Table 10 reveal that increasing N improves PSNR, indicating that finer temporal resolution enhances reconstruction quality. However, this improvement comes at the cost of an extended training time, increasing from 1.3 hours at N=2 to 7 hours at N=6. Importantly, our method maintains real-time rendering performance (30+ FPS), matching CF-3DGS as both utilize the same efficient 3DGS rendering framework. This gives our approach a significant advantage over slower NeRF-based methods. Even at N=2, our approach outperforms baselines in both PSNR and rendering speed. Event-3DGS achieves a good balance of efficiency and performance, but it requires precomputed poses. The setting of N=3 offers a good balance, achieving higher PSNR than CF-3DGS with comparable training time. However, it's important to note that the optimal N may vary for different frame rates and motion characteristics.

Table 10: Effect of the number of event subintervals N.

| | PSNR↑ | FPS↑ | t↓ |
|---|---|---|---|
| $N = 2$ | 23.56 | 30+ | 1.3h |
| $N = 3$ | 24.43 | 30+ | 3h |
| $N = 6$ | **24.81** | 30+ | 7h |
| LocalRF | 21.20 | <1 | 8h |
| CF-3DGS | 22.08 | 30+ | 3h |

Figure 8: Comparison of training time vs PSNR for various methods

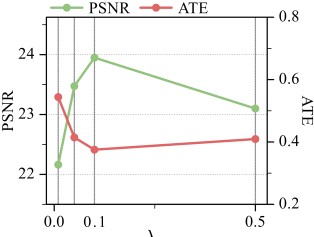

Figure 9: Study of the different evaluation metrics with respect to $\lambda_{cm}$

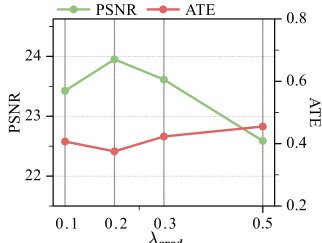

Figure 10: Study of the different evaluation metrics with respect to $\lambda_{grad}$

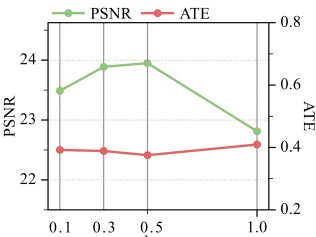

Figure 11: Study of the different evaluation metrics with respect to $\lambda_{PBA}$

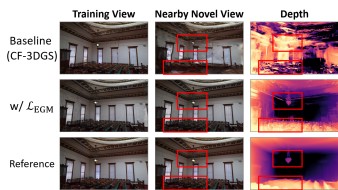

Figure 12: Qualitative Results of the effectiveness of $\mathcal{L}_{EGM}$.

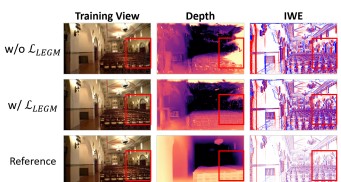

Figure 13: Qualitative Results of the effectiveness of $\mathcal{L}_{LEGM}$.

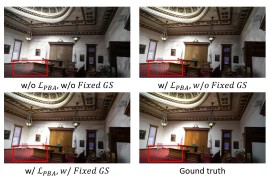

Figure 14: Qualitative Results of the effectiveness of $\mathcal{L}_{PBA}$.

#### A.4.4 Analysis of Loss Coefficients

Different components of our method play crucial roles in overall performance. To comprehensively understand these effects and determine optimal weight settings, we conducted a detailed analysis of the loss weights: $\lambda_{cm}$, $\lambda_{grad}$ and $\lambda_{PBA}$.

**Contrast Maximization Coefficient $\lambda_{cm}$** As shown in Fig. 9, the contrast maximization coefficient $\lambda_{cm}$ significantly affects both NVS quality and pose estimation. We observe that a value of 0.1 achieves the best overall performance. Lower values (0.01) lead to decreased performance, likely due to insufficient utilization of motion information from events. Higher values (0.5) also result in performance degradation, possibly due to over-reliance on the contrast maximization term at the expense of other constraints. The optimal $\lambda_{cm}$ suggests that while event data is crucial, it should not dominate the reconstruction process. This balance allows our method to leverage the high temporal resolution of events without sacrificing the global supervision provided by frame data.

**Gradient-based Loss Coefficient $\lambda_{grad}$** As shown in Fig. 10, the gradient-based loss coefficient $\lambda_{grad}$ shows optimal performance at 0.2. Lower values (0.1) slightly decrease performance, while higher values (0.3, 0.5) lead to more significant drops in both NVS quality and pose accuracy. This is likely because $\mathcal{L}_{EGM}$ and 3DGS primarily focus on rendering absolute brightness, and excessive gradient constraints may interfere with this process. Therefore, a moderate $\lambda_{grad}$ value is crucial to balance gradient information with the primary rendering objectives.

**Photometric Bundle Adjustment Coefficient $\lambda_{PBA}$** As illustrated in Figure 11, our method demonstrates remarkable stability across $\lambda_{PBA}$ values ranging from 0.1 to 0.5. However, while $\mathcal{L}_{PBA}$ has minimal effect on PSNR and ATE metrics, its absence leads to noticeable color distortion artifacts in visual results. This underscores the importance of $\lambda_{PBA}$ in maintaining visual fidelity. Conversely, setting $\lambda_{PBA}$ to 1.0 results in significant performance degradation, indicating that overemphasis on cross-view consistency can be counterproductive.

### A.5 Visualization of Ablation Studies

Due to space constraints, we provide comprehensive visualizations of our ablation studies in this section. Our visualizations demonstrate that while CF-3DGS can produce high-quality renderings at training views, it struggles with adjacent novel views. The rendered depth maps expose considerable inaccuracies in scene geometry reconstruction. With $\mathcal{L}_{EGM}$, as shown in Fig. 12, we observe marked improvements as it effectively harnesses continuous brightness change data captured by event cameras, compensating for inter-frame information loss. For $\mathcal{L}_{LEGM}$, as shown in Fig. 13, our visualizations show that without this component, the generated Image of Warped Events (IWE) exhibits blurring effects, indicating inaccurate pose estimation. This is particularly evident in self-similar regions with repetitive textures, which pose additional challenges. $\mathcal{L}_{LEGM}$ optimizes IWE sharpness, effectively improving both rendering quality and pose estimation. Regarding $\mathcal{L}_{PBA}$ and Fixed-GS, as shown in Fig. 14, our visualizations reveal that without these components, the reconstructed scene loses almost all color information. This occurs because event-guided optimization, with substantially more event frames than image frames, dominates scene reconstruction and suppresses color information. $\mathcal{L}_{PBA}$ alone, which reprojects neighboring video frames onto event frames, only partially addresses this imbalance. The Fixed-GS strategy separates color and structure optimization, and when combined with PBA, they complement each other to effectively resolve color distortion issues.

### A.6 Additional Challenging Scenario Experiments

To further demonstrate event camera advantages, we constructed a FASTER scenario by selecting every 10th frame from the SLOW dataset and directly applied COLMAP pose estimation on these challenging sequences. We added COLMAP+3DGS as a comparison baseline. Notably, in the corner and outdoor scenes, several frames exhibit registration failures where the estimated poses greatly deviate from the overall video trajectory, indicating COLMAP's limitations in challenging real-world scenarios.

Table 11: Rendering quality comparison across different speed scenarios on RealEv-DAVIS dataset.

| Methods | SLOW | | FAST | | FASTER | |
|---|---|---|---|---|---|---|
| | PSNR↑ | SSIM↑ | PSNR↑ | SSIM↑ | PSNR↑ | SSIM↑ |
| CF-3DGS | 22.68 | 0.629 | 17.59 | 0.520 | 15.61 | 0.504 |
| COLMAP+3DGS | 23.27 | 0.633 | 19.79 | 0.548 | 17.25 | 0.521 |
| EF-3DGS (Ours) | **23.65** | **0.647** | **21.12** | **0.562** | **20.18** | **0.532** |

As shown in Table 11, the results demonstrate that: (1) COLMAP+3DGS rendering quality degrades significantly as motion speed increase confirming pose estimation inaccuracies in challenging conditions. (2) Compared to CF-3DGS, our method maintains substantially better rendering quality in high-speed scenarios, highlighting our method robustness and demonstrating the potential of event cameras in high-speed scenarios.

### A.7 Necessity of Two-Stage Optimization

To validate the necessity of our two-stage Fixed-GS training strategy, we explore an alternative approach: incorporating a reference view rendering loss $\mathcal{L}_{\text{ref}}$ at each training iteration to address the color distortion challenge without requiring a separate optimization stage.

Given that events have much higher temporal resolution than RGB frames, the sparse color constraints from RGB images are largely overwhelmed by the abundant grayscale constraints from event data, ultimately resulting in color distortion. By incorporating $\mathcal{L}_{\text{ref}}$, the backpropagated gradients at each iteration would be augmented with color information from the reference view, potentially alleviating the color distortion issue. We conducted additional experiments comparing this approach with our two-stage strategy. As shown in Table 12, the results show that incorporating $\mathcal{L}_{\text{ref}}$ achieves comparable rendering quality. Although we cannot include visualizations here, our qualitative results confirm that the color distortion issue is effectively alleviated with this approach. However, this comes at a significant computational cost, increasing training time (3.8h vs 2.5h) due to additional reference view rendering at each iteration. Considering the modest performance difference relative to the substantial training overhead, we believe the two-stage optimization strategy offers a more efficient and practical solution.

Table 12: Comparison of different color correction strategies.

| Method Description | PSNR↑ | SSIM↑ | LPIPS↓ | Training Time↓ |
|---|---|---|---|---|
| Baseline | 23.09 | 0.70 | 0.38 | 2.7h |
| $\mathcal{L}_{event} + \mathcal{L}_{color}$ | **24.11** | **0.73** | **0.36** | 3.8h |
| $\mathcal{L}_{event}$ + two-stage (Fixed-GS) strategy | 23.96 | 0.72 | **0.36** | **2.5h** |

### A.8 Additional Visualization

We present additional qualitative results on both Tanks and Temples and RealEv-DAVIS.

### A.9 Limitations

Our method combines event streams with conventional video frames, which require the input data to be time-ordered. So it can not handle the unordered image input. And, our method relies on several

key parameters (e.g., $\lambda_{cm}$, $\lambda_{grad}$ and $\lambda_{PBA}$) that may require manual tuning for optimal performance across different speed scenarios. This dependency on scene-specific parameter settings could limit the method's adaptability to diverse environments. Future work should explore self-adaptive parameter adjustment strategies to enhance the method's versatility and ease of use across various reconstruction tasks.

## A.10   Broader Impacts

To the best of our knowledge, the proposed method will not have significant negative social impact. The proposed scene reconstruction method can be used to reconstruct and render some wild scenes. Users can use the video shot by their mobile phones as input to obtain an explicit 3D asset represented by a 3D Gaussian. This 3D asset can be used for subsequent editing, development, secondary creation for entertainment.

## A.11   Data Availability

The datasets that support the findings of this study are available in the following repositories: Tanks and Temples [50] at `https://www.tanksandtemples.org/` under CC BY 4.0 license, The code of choosen baseline, Nope-NeRF [10] is available at `https://github.com/WU-CVGL/BAD-Gaussians` under Apache-2.0 license, LocalRF [8] is available at `https://localrf.github.io/` under Apache-2.0 license, Event-3DGS [44] is available at `https://github.com/lanpokn/Event-3DGS` under Apache-2.0 license, EvDeblurNeRF [40] is available at `https://github.com/uzh-rpg/EvDeblurNeRF` under Apache-2.0 license, ENeRF [36] is available at `https://github.com/knelk/enerf` under MIT license, CF-3DGS [5] is available at `https://github.com/NVlabs/CF-3DGS` under Apache-2.0 license, .

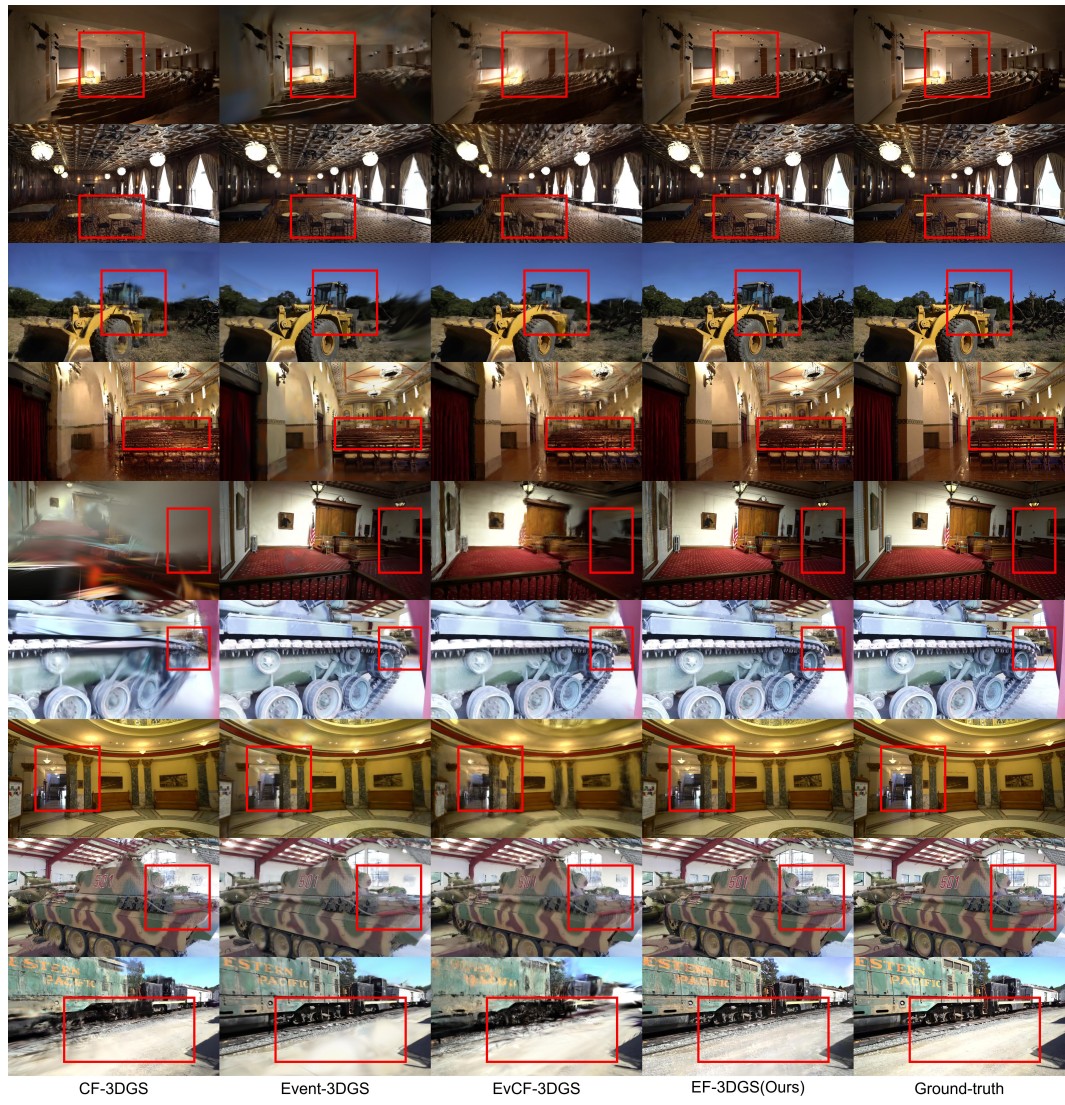

| CF-3DGS | Event-3DGS | EvCF-3DGS | EF-3DGS(Ours) | Ground-truth |

Figure 15: Qualitative comparison for novel view synthesis on Tanks and Temples. Our approach produces more realistic rendering results with fine-grained details. Better viewed when zoomed in.

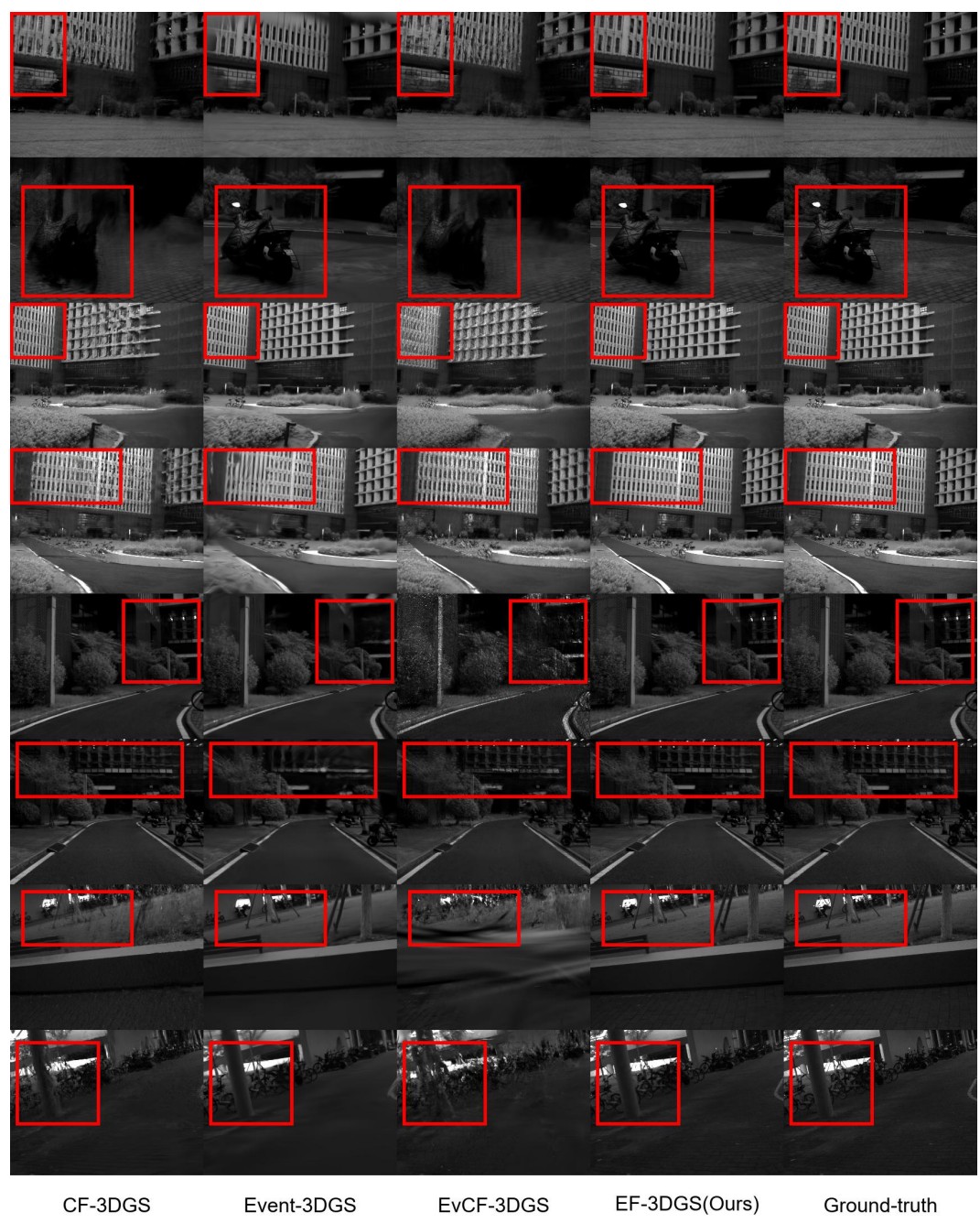

|  CF-3DGS | Event-3DGS | EvCF-3DGS | EF-3DGS(Ours) | Ground-truth |

Figure 16: Qualitative comparison for novel view synthesis on RealEv-DAVIS. Our approach produces more realistic rendering results with fine-grained details. Better viewed when zoomed in.

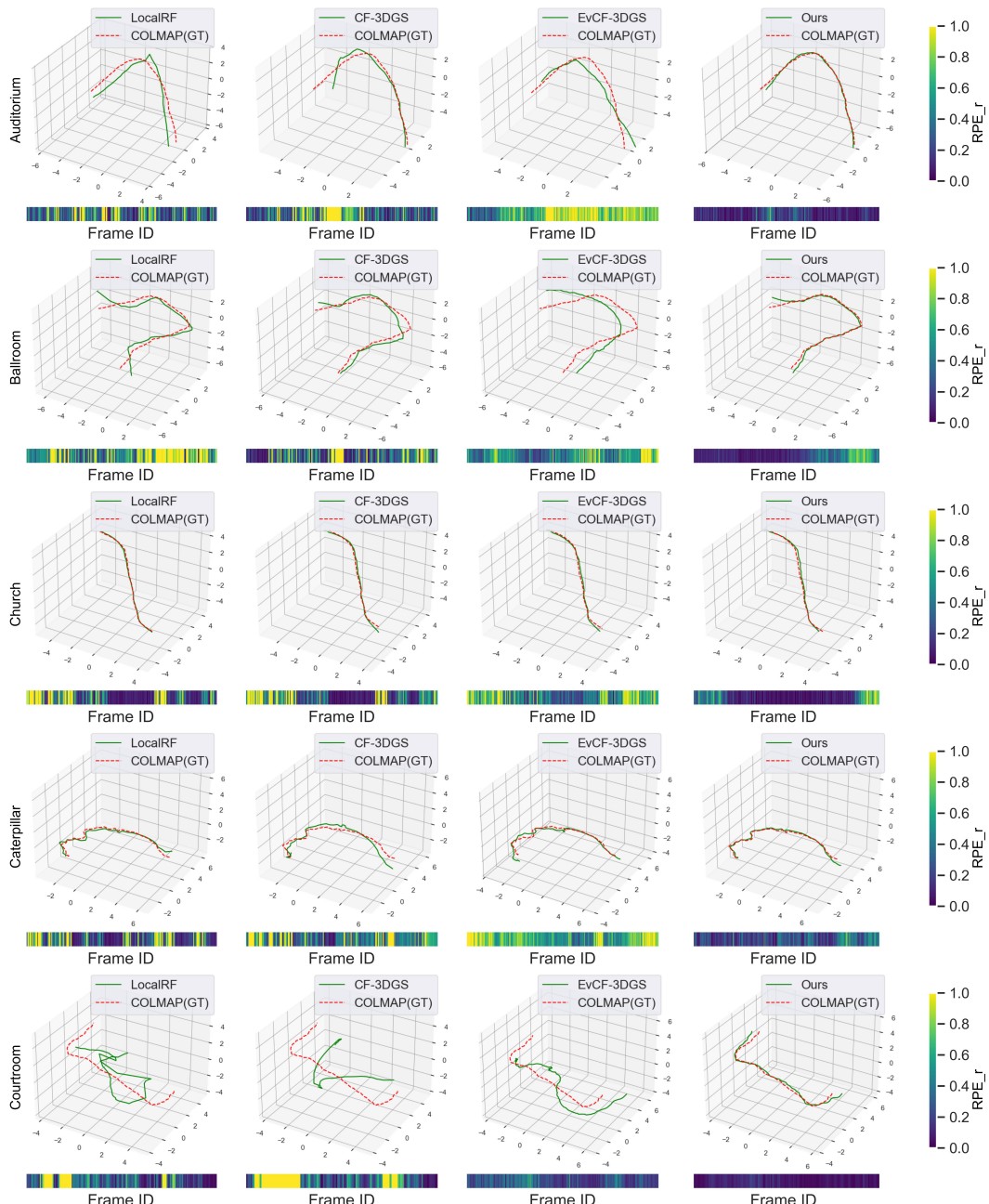

Figure 17: Pose Estimation Comparison on Tanks and Temples. We visualise the trajectory (3D plot) and relative rotation errors RPE$_r$ (bottom colour bar) of each method. We clip and normalize the RPE$_r$ by a quarter of the max RPE$_r$ across all results of each scene.

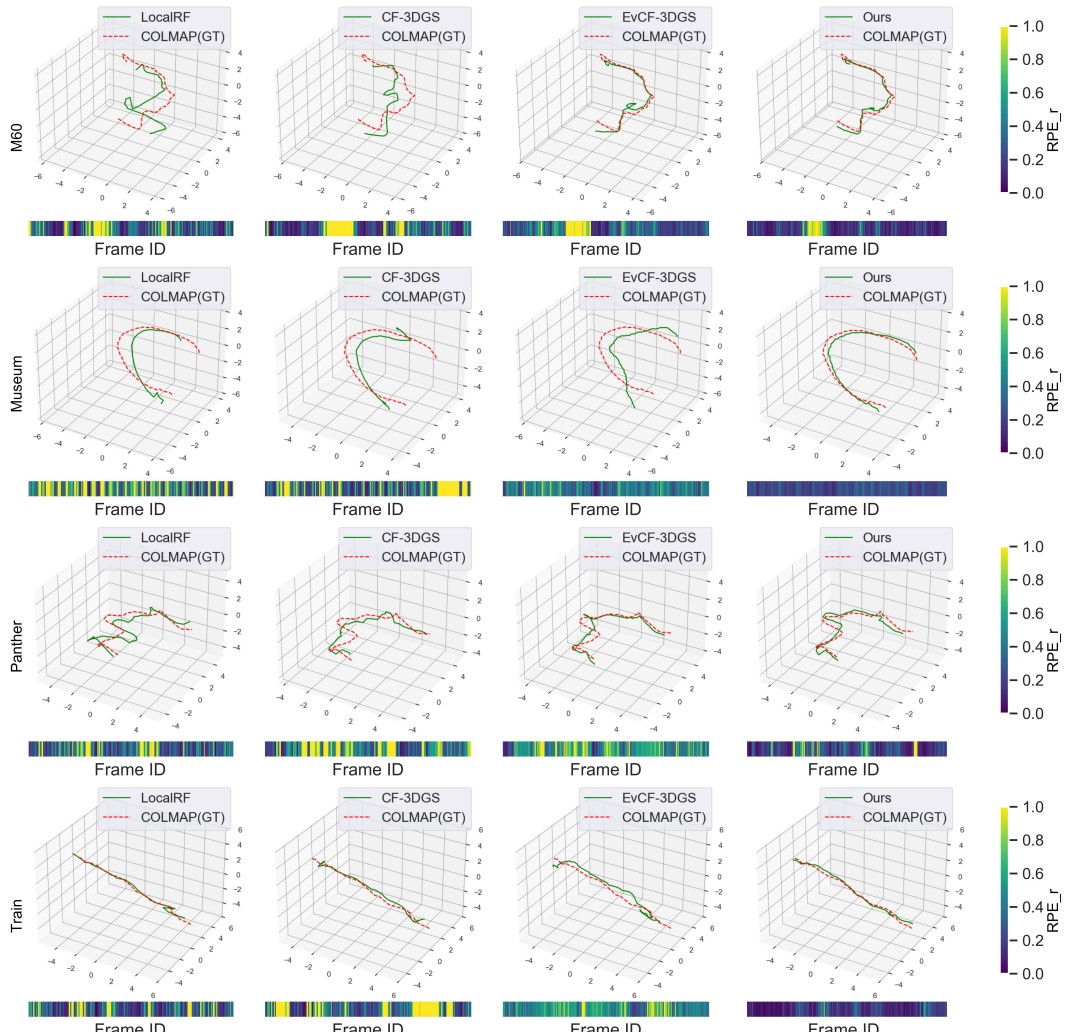

Figure 18: Pose Estimation Comparison on Tanks and Temples. We visualise the trajectory (3D plot) and relative rotation errors $RPE_r$ (bottom colour bar) of each method. We clip and normalize the $RPE_r$ by a quarter of the max $RPE_r$ across all results of each scene.

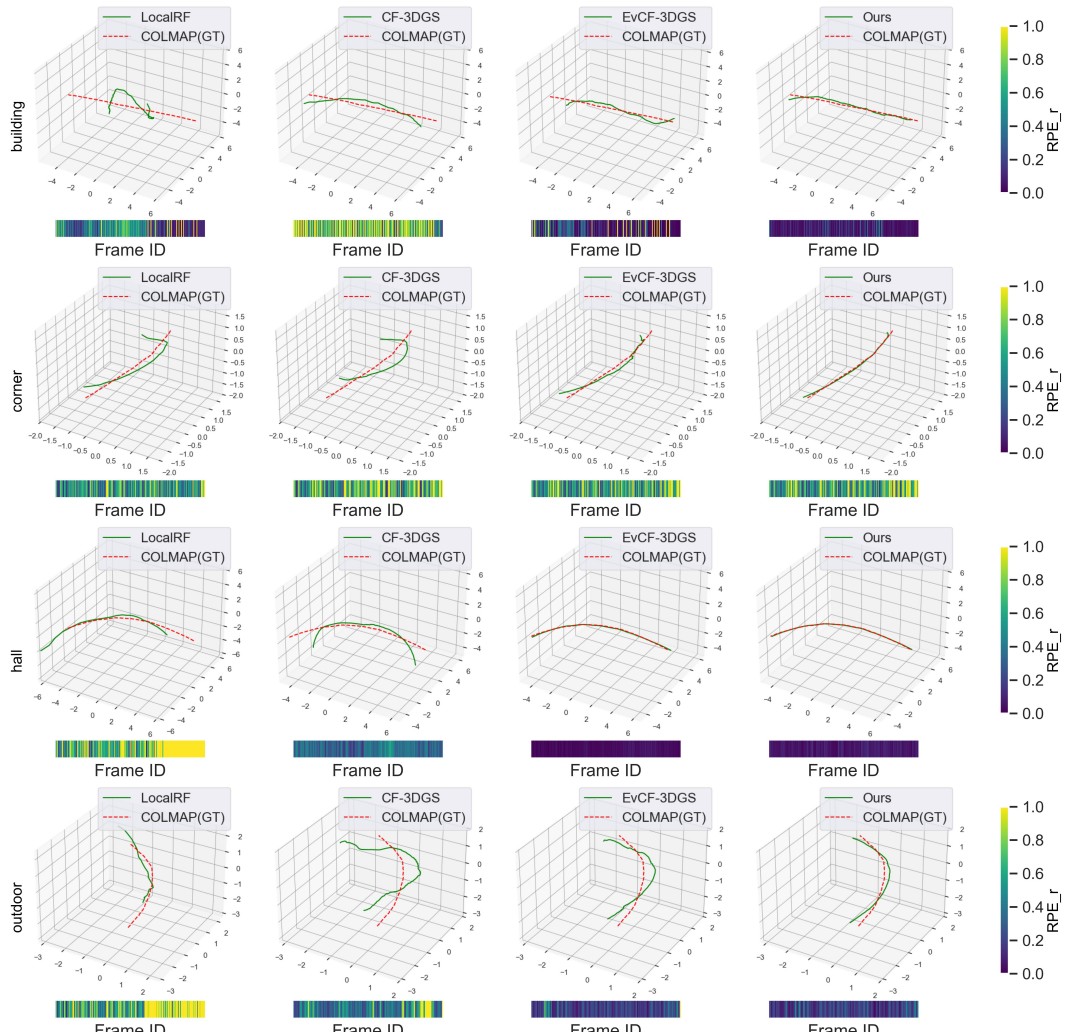

Figure 19: Pose Estimation Comparison on Tanks and Temples and RealEv-DAVIS. We visualise the trajectory (3D plot) and relative rotation errors $RPE_r$ (bottom colour bar) of each method. We clip and normalize the $RPE_r$ by a quarter of the max $RPE_r$ across all results of each scene.

