# OpenReview forum: "EF-3DGS: Event-Aided Free-Trajectory 3D Gaussian Splatting"
_NeurIPS.cc/2025/Conference — NeurIPS 2025 spotlight_

### Official Review · Reviewer_rd3s · 2025-06-15

**Clarity:** 4
**Significance:** 3
**Originality:** 3
**Rating:** 5
**Confidence:** 4

**Summary:**

This paper, EF-3DGS: Event-Aided Free-Trajectory 3D Gaussian Splatting, proposes the first method combining event cameras with 3D Gaussian Splatting (3DGS) without requiring pre-estimated camera poses. The approach jointly optimizes camera poses and Gaussian positions using event data, while leveraging image data for color optimization. The method employs a two-stage training pipeline and is thoroughly validated through extensive experiments.

**Questions:**

### Questions about limitations
**1. On the sharpness of pose optimization (related to L169):**
The method appears to encourage maximizing alignment sharpness via event warping. However, event triggering is itself subject to latency and noise, especially under low-light conditions. Could this lead to overfitting to noise, i.e., aligning events that should not be aligned in the first place? If so, does this affect the robustness of the method under high noise or delayed event response?

**2. On gradient-based loss (related to L189):**
Event cameras operate asynchronously on a per-pixel basis. Does converting a series of events into a difference between two pixels rely on overly strict assumptions about the scene? Only assuming static scenes and consistent lighting seems not enough. For instance, if a light source is partially occluded while moving, or if surface materials exhibit strong anisotropy, could these factors lead to significant degradation in reconstruction quality?

**3. On stability:**
Event data is inherently noisy, especially under low lighting or with sensor latency. Moreover, 3D Gaussian Splatting with difference-based loss can be fragile . The proposed method appears not to include explicit noise-handling mechanisms. Given these factors, how does the approach achieve stable reconstruction? Were there any architectural or algorithmic choices that implicitly contribute to robustness?

---


### Other minor questions

**4. Regarding the definition of $\Delta L$ (L178):**
It is unclear what the exact meaning of $\Delta L$ is — is it the per-location brightness change? If so, the right-hand side of the formula might effectively reduce to the number of events. Could the authors clarify?

**5. On dataset realism and evaluation settings (L249):**
The use of COLMAP to obtain ground truth poses suggests the dataset is relatively ideal (i.e., low speed and good lighting). Are there any results — even qualitative — under high-speed or low-light conditions, where event cameras typically shine? Such evaluations would strengthen the claim that the method is robust and practically applicable.

---
Due to some remaining unresolved questions, my current recommendation is borderline accept. Nevertheless, I expect to raise my evaluation after reviewing the authors’ rebuttal.

**Ethical Concerns:**

["NO or VERY MINOR ethics concerns only"]

**Final Justification:**

The authors’ rebuttal is very thorough and thoughtfully addresses most of my concerns. Combined with the fact that the original manuscript is of good quality, I have decided to raise my score accordingly.

**Limitations:**

Some important limitations are not adequately discussed. My concerns are already outlined in the "Questions about limitations" section of this review. I look forward to the authors’ reflections on these points.

**Paper Formatting Concerns:**

No major formatting issues noticed.

**Quality:**

4

**Strengths And Weaknesses:**

Strengths:
- The paper is very well written, with clear logical flow and well-organized presentation, making it easy to follow and understand.
- The theoretical formulation is clearly explained, and the authors properly acknowledge the inspiration and connections to prior work.
- The experiments are thorough and comprehensive, covering ablation study.

Weaknesses:
- The discussion on the method's limitations may not be sufficiently comprehensive. The current design may restrict its applicability to certain environments.
- The experimental setup lacks scenarios involving high-speed motion or low-light conditions, which are typically where event cameras provide the greatest advantage. This limits the demonstration of the full potential of combining event data with 3DGS.

---

> ### Author Rebuttal · Authors · 2025-07-31
>
> We thank the reviewer for the constructive feedback and for recognizing our well-written paper, clearly explained formulation, and comprehensive experimental validation. We respond to each concern in detail. Please feel free to use the discussion period if you have any additional questions.
>
> > ### **Q1: Questions on the Sharpness of Pose Optimization**
>
> **Answer:** We understand the reviewer's concern regarding how event noise could lead to erroneous warping. We address this question from two perspectives:
>
> **Robustness Discussion:** Thank you for this insightful question. Indeed, the event-based vision community has extensively studied misalignment issues in event warping [1,2]. Misalignment primarily stems from two factors: 1) event noise [1] and 2) the ill-posed nature of the alignment optimization process (existence of local minima) [2], with these two factors often interfering with each other. While event noise inevitably causes alignment difficulties, we believe our method demonstrates certain noise robustness capabilities. In our approach, we derive the motion field from 3DGS-rendered depth and inter-frame poses, then use this motion field to warp events. This formulation involves optimizing two sets of parameters: 3DGS-rendered depth (corresponding to scene reconstruction) and camera poses (corresponding to camera trajectory estimation), creating a joint pose estimation and scene reconstruction problem (chicken-and-egg problem). Our method begins with monocular depth estimation networks to obtain a reasonable 3DGS initialization. With this initially good scene representation, the solution space is significantly reduced. Specifically, noise events typically lack spatial consistency, while our 3DGS depth-based motion field requires spatial geometric consistency, making our optimization inherently robust to isolated noise events. This constraint allows us to primarily optimize pose parameters (6DOF), making the maximization of IWE sharpness more robust, even in the presence of noise.
>
> **Noise Experiment Results:** To investigate the impact of noise on our method, we conduct experiment on Tanks and Temples and, following [2], randomly added noise events to the event stream based on a certain percentage of the total event count. The experimental results are shown below. Within a reasonable range (up to 30%), our method demonstrates considerable noise robustness, with pose estimation accuracy degrading gracefully rather than catastrophically.
>
> | Noise Level | PSNR↑ | SSIM↑   | $RPE_t$↓ | $RPE_r$↓  |
> |-|--|--|-|----|
> | 0% | 23.96 | 0.72  | 0.049 | 0.626 |
> | 10% | 23.28 | 0.71  | 0.064 | 0.661 |
> | 30% | 22.16 | 0.69  | 0.088 | 0.727 |
> | 50% | 20.24 | 0.65  | 0.125 | 0.854 |
>
> **Summary:** We share our thoughts on how noise affects event warping alignment and conducted preliminary noise experiments. While our primary contribution focuses on demonstrating robustness across different frame rates (high-speed scenarios), we acknowledge that comprehensive noise analysis requires further investigation in challenging real-world conditions. We find noise robustness to be a fascinating research direction and are willing to incorporate this discussion in the final version.
>
> > ### **Q2: Questions on the Gradient-Based Loss**
>
> **Answer:** We thank the reviewer for this insightful question. The gradient-based loss indeed relies on the brightness constancy assumption $L(x,y,t) = L(x+\Delta x, y+\Delta y, t+\Delta t)$, which, combined with the small motion assumption, establishes the relationship between brightness changes, image gradients, and optical flow as shown in Eq. 11: $ \Delta L(u) = - \nabla L \cdot \dot{u}$. In typical scenarios, these assumptions generally hold over short time intervals thanks to the high temporal resolution characteristics of event cameras. The challenging scenarios mentioned by the reviewer (moving light occlusion and strong material anisotropy) do violate this assumption. However, this limitation is not unique to our event-based approach but represents a fundamental challenge shared across photometry-based reconstruction methods. As acknowledged in [3], existing event-based intensity reconstruction methods face the same constraint. Moreover, traditional 3DGS itself relies heavily on cross-view photometric consistency to establish geometric constraints—the scenarios described by the reviewer would equally compromise this consistency and degrade reconstruction quality in conventional 3DGS pipelines. Addressing such complex light-material interactions would require explicit modeling of surface material properties and illumination dynamics, which falls within the scope of inverse rendering rather than geometric reconstruction. We believe leveraging event data for inverse rendering represents an exciting future research direction that merits dedicated investigation.
>
> > ### **Q3: Questions on the Stability**
>
> **Answer:** Thank you for raising this important stability concern. For experiments evaluating our robustness, please refer to **Noise Experiment Results** in **Q1**. We would like to further explain our method's robustness:
>
> **1) Our Distinction from Other Event-Based NeRF or 3DGS Methods:** Previous event-based reconstruction methods EventNeRF and Event3DGS typically render frames at two different timestamps, compute the logarithmic brightness difference between these frames, and compare it with accumulated events between the two timestamps to calculate the loss. As the reviewer points out, computing differences between two images is inherently unstable, as differential operations further amplify noise and training instability. Moreover, events naturally record brightness changes rather than absolute intensities, making them unsuitable for rendering. Previous methods treat events as a **substitute for RGB images**, whereas we treat events as **auxiliary modal data to complement RGB images**. Through EGM and LEGM, we simultaneously leverage both brightness and brightness changes to assist RGB images in scene reconstruction, providing complementary supervision rather than replacement. Meanwhile, as camera motion speed increases (reflected by decreasing frame rates in our experiments), the potential of event data becomes increasingly prominent.
>
> **2) Robustness Potential of $L\_{grad}$:** According to our derivation and Equation 9:
> $$IPWE\_{i,j} = \frac{1}{r+1}\sum\_{m=j-r}^{j}E\_{i,m \rightarrow j}$$
> For each warped event $E\_{i,m-j}$, we model it using a simple i.i.d. Gaussian noise model:
> $$
> E\_{i,m-j} = \frac{1}{C}\Delta L + n\_{m \rightarrow j},
> $$
> where $n_{m \rightarrow j} \sim N(0, \sigma^2)$ follows a Gaussian distribution. Therefore:
> $$IPWE\_{i,j} = \frac{1}{r+1}\sum\_{m=j-r}^{j}E\_{i,m \rightarrow j} = \frac{1}{C}\Delta L + \frac{1}{r+1}\sum\_{m=j-r}^{j} n\_{m \rightarrow j}. $$
> The variance of the noise term  $\frac{1}{r+1}\sum\_{m=j-r}^{j} n\_{m \rightarrow j}$ is  $\frac{\sigma^2}{r+1}$, which demonstrates that averaging warped events significantly reduces noise variance by a factor of $(r+1)$. This averaging mechanism inherently provides noise suppression, making our $L_{\text{grad}}$ more robust compared to methods that rely on single-frame differences. While this derivation may have some simplifications, we hope this example illustrates the robustness potential of $L_{\text{grad}}$.
>
> > ### **Q4: Questions Regarding the Definition of $\Delta L$**
>
> **Answer:** Yes, $\Delta L$ represents the logarithmic brightness change. Each term $E\_{i,m \rightarrow j}$ on the right-hand side of the equation represents the accumulated events within the interval that are warped to the target timestamp $j$.
>
> > ### **Q5: Questions On Dataset Realism and Evaluation Settings**
>
> **Answer:** We thank the reviewer for this important question regarding dataset realism and challenging conditions evaluation. We would like to refer to our comprehensive response in **Quality 2: Ground Truth Pose Concerns and Challenging Scenario Experiments** from reviewer **UuWp's** reviews, where we have extensively addressed similar concerns about high-speed and challenging conditions.
>
> In the **RealEv-DAVIS Ground Truth Pose Clarification** section, we explained our dataset's ground truth pose acquisition process which ensures fair quantitative comparison while simulating realistic challenging conditions.
>
> In the **Additional Challenging Scenario Experiments** section, we conducted supplementary experiments under FASTER scenarios on RealEv-DAVIS where COLMAP frequently fails due to insufficient visual overlap and rapid motion. These experiments demonstrate that our event-aided method maintains superior performance compared to traditional approaches even under extreme challenging conditions.
>
> As further elaborated in our response to reviewer **hsxP**'s **Justification for Pose-Free Approach** weakness, our novelty lies in effectively leveraging event camera properties to address two fundamental challenges that existing pose-free methods face in high-speed scenarios: (1) sparse scene reconstruction due to insufficient frame observations, and (2) inaccurate pose estimation caused by large inter-frame displacements.
>
>
> [1] Stoffregen, Timo, and Lindsay Kleeman. "Event cameras, contrast maximization and reward functions: An analysis." Proceedings of the IEEE/CVF Conference on Computer Vision and Pattern Recognition. 2019.
>
> [2] Shiba, Shintaro, Yoshimitsu Aoki, and Guillermo Gallego. "Secrets of event-based optical flow." European Conference on Computer Vision. Cham: Springer Nature Switzerland, 2022.
>
> [3] Zhang, Zelin, Anthony J. Yezzi, and Guillermo Gallego. "Formulating event-based image reconstruction as a linear inverse problem with deep regularization using optical flow." IEEE Transactions on Pattern Analysis and Machine Intelligence 45.7 (2022): 8372-8389.

---

> > ### Comment · Reviewer_rd3s · 2025-08-04
> >
> > Thank you for your insightful and detailed response. I appreciate the clarifications provided and have updated my score accordingly.

---

> ### Author Response · Authors · 2025-08-04
> **Thanks to Reviewer rd3s**
>
> Dear Reviewer rd3s,
>
> Thank you for your thoughtful consideration and constructive feedback on our work! We are truly grateful for your recognition of our contributions and deeply appreciate the insightful comments regarding the robustness of our proposed method. Your suggestions have prompted us to engage in deeper reflection on our approach, which not only helped us address your concerns but also led us to develop additional insights that strengthen the robustness of our method. We are committed to incorporating your valuable recommendations into the final version of our manuscript, specifically enhancing our analysis of the method's robustness while presenting these new perspectives that emerged from our discussions.
>
> Once again, we sincerely appreciate your expertise and the time you have dedicated to reviewing our submission : )
>
> Best regards,
>
> Authors of Submission 21739

---

### Official Review · Reviewer_UuWp · 2025-06-25

**Clarity:** 3
**Significance:** 2
**Originality:** 3
**Rating:** 5
**Confidence:** 4

**Summary:**

This paper proposes a novel method for 3D Gaussian Splatting of static scenes using images and event streams as input, while simultaneously optimizing the camera trajectory. The input images are temporally sparse, whereas the events provide dense temporal information. The proposed method consists of two stages. In the first stage, the parameters of the 3D Gaussians and the camera trajectory are jointly optimized using a loss function derived from the Event Generation Model (EGM), Contrast Maximization (CMax), and Photometric Bundle Adjustment (PBA). Since events do not contain information about absolute brightness values, accurate color reconstruction is not feasible in this stage. Therefore, in the second stage, only the color of the 3D Gaussians are refined using the RGB images.

In the experiments, the authors used the publicly available Tanks and Temples dataset, as well as the RealEv-DAVIS dataset constructed by the authors, to quantitatively and qualitatively evaluate the effectiveness of the proposed method. Additionally, the authors assessed the robustness of the proposed method to noise in the initial camera poses and demonstrated that it maintains strong performance under high noise levels compared to existing approaches.

**Questions:**

1. Contrast Maximization optimization is generally computationally expensive. In this paper, instead of computing the Image of Warped Events (IWE) for each individual event, the authors propose computing the Image of Piece-wise Warped Events (IPWE) by warping events that occur within a predefined time interval collectively. How much does this modification reduce the overall optimization time?
2. In Section 4.4, the authors state, “The ratio between the first and second stages is empirically set to 4:1.” What does this ratio refers to? Does it indicate the ratio of the number of iterations used in the optimization for each stage?
3. What is Ev-Baseline in Table 8?
4. How is the initial camera poses given?

**Ethical Concerns:**

["NO or VERY MINOR ethics concerns only"]

**Final Justification:**

The authors provided sufficient experimental results in their rebuttal, which led me to revise my previous rating.
I believe that if the results of the experiments conducted by the authors in their rebuttal are incorporated into the final version of the paper, it will be worthy of acceptance by NeurIPS.

**Limitations:**

yes.

**Paper Formatting Concerns:**

no.

**Quality:**

2

**Strengths And Weaknesses:**

**Quality**
1. Questions remain regarding the experimental assumptions. The authors simulate large inter-frame displacements by reducing the number of frames. However, high-speed camera motions would typically result in motion blur in the captured images. The experiments in the paper assume that sharp, blur-free images are provided as input, which may not accurately reflect real-world conditions.
2. In the RealEv-DAVIS dataset, COLMAP is used to obtain ground-truth camera poses, which implies that accurate camera pose can be estimated from images. To more clearly demonstrate the advantages of event cameras, it may be beneficial to conduct additional experiments where pose estimation from images is challenging or unreliable.
3. Although the experiments in this paper assume that the threshold $C$ at which events are triggered is known, this assumption may not hold in general.

**Clarity**
1. The process of the proposed method is very clearly explained in the paper and is easily understood.
2. The authors present extensive quantitative and qualitative experimental results, which effectively demonstrate the efficacy of the proposed method and aid the reader's understanding.

**Significance**
1. This paper serves as an example demonstrating the advantages of event cameras over frame-based cameras in 3D Gaussian Splatting. However, as previously noted, questions remain regarding the experimental setup, and the full potential of event cameras has not been comprehensively evaluated.

**Originality**
1. The authors leverage the unique characteristics of events by integrating frame and image information through the EGM, and by utilizing motion information encoded in the events for optimization via Contrast Maximization.

---

> ### Author Rebuttal · Authors · 2025-07-31
>
> We thank the reviewer for the constructive feedback. We respond to each concern in detail. Please feel free to use the discussion period if you have any additional questions.
>
> > ### **Quality 1: Questions Regarding the Experiment Assumptions.**
>
> **Answer:** We sincerely appreciate the reviewer's insightful observation regarding motion blur. We respond to this concern from two perspectives:
>
> **1) Discussion on Motion Blur**: We acknowledge that motion blur and large pixel displacements may simultaneously occur in high-speed scenarios. To address the reviewer's concern about motion blur in our experimental setup, we would like to clarify our experimental setup and novel contribution through an established paradigm in computer vision.
>
> In low-level vision, image deblurring and video frame interpolation represent fundamentally different challenges: image deblurring addresses degradation **within individual frames** during exposure time, while video frame interpolation tackles discontinuity **between different frames** due to large pixel displacements. Notably, video frame interpolation methods typically do not consider motion blur in their evaluation.
>
> Drawing from this paradigm, we can view 3D reconstruction similarly: our work specifically targets sparse viewpoint and pose estimation challenges arising from large pixel displacements between frames, while motion blur represents intra-frame degradation—a separate problem domain. Previous event-based 3D methods have focused on **intra-frame** blur mitigation assuming accurate poses. In contrast, we address fundamental **inter-frame** challenges, specifically joint pose-scene optimization and sparse view reconstruction, which are fundamental robustness challenges in 3DGS reconstruction.
>
>
> **2) Additional Experimental Validation**: For completeness, we conducted additional experiments to evaluate our method's performance under motion blur conditions. We extend our method using the Event Double Integration (EDI) model[1] by reformulating the intensity term $I\_{i,0}$ in Eq. (5) using the EDI formulation:
> $$
> \hat{I}\_{i,0} = \frac{(2n + 1) \cdot B\_i}{\sum_{k=-n}^{n} \exp\left(C \cdot \sum\_{z=0}^{k} E\_{i,z}\right)},
> $$
> where $B_i$ represents the blurred intensity, $E_{i,z}$ denotes the accumulated events, $C$ is the contrast threshold, and $n$ defines the temporal integration window of blur averaging.
>
> Experiments were performed on the Tanks and Temples dataset with a frame rate of 2FPS. Motion-blurred frames were synthesized adopting gamma correction and multi-frame averaging operations. We average every 30 frames to simulate blurring. We selected EvDeblurNeRF[2], currently the best-performing open-source event-based deblur scene reconstruction method, as our comparison.
>
> | Methods | Pose Estimation | PSNR↑ | SSIM↑   | $RPE_t$ ↓ | $RPE_r$ ↓ |
> |-|-|-|-|-|-|
> | EvDeblurNeRF  | CF-3DGS | 21.17 | 0.69  | 0.105 | 0.897 |
> | EvDeblurNeRF  | COLMAP | 22.58 | 0.70  | - | - |
> | EDI + EF-3DGS (Ours) | EDI + EF-3DGS (Ours) | **23.18** | **0.72**  | **0.062** | **0.642** |
>
> As demonstrated in the table, our method achieves superior reconstruction quality to EvDeblurNeRF while simultaneously performing better pose estimation. Note that EvDeblurNeRF requires pre-computed COLMAP poses, which significantly restricts practical applicability.
>
> **Summary**: We appreciate the reviewer's concern about motion blur. We clarify that our method focuses on large pixel displacement challenges, which represents a complementary challenge to intra-frame motion blur. Our additional blur experiments suggest that our approach can be reasonably extended to handle motion blur scenarios. We acknowledge that more extensive realistic evaluation under various blur conditions would strengthen our claims, and we commit to conducting comprehensive validation with real-world motion blur and diverse challenging scenarios in the revised manuscript.
>
> > ### **Quality 2: Ground Truth Pose Concerns and Challenging Scenario Experiments**
>
> **Answer:** We sincerely thank the reviewer for this important suggestion. We would like to clarify the experimental setup and provide additional validation.
>
> **RealEv-DAVIS Ground Truth Pose Clarification:** In the SLOW scenario, COLMAP provides relatively accurate pose estimation due to sufficient visual overlap and features. The FAST scenario ground truth poses are then obtained by downsampling from these accurate SLOW scenario COLMAP poses, rather than directly running COLMAP on the sparse FAST frames. This enables fair quantitative comparison of pose estimation accuracy between methods. Our experiment evaluation protocol follows previous pose-free reconstruction methods, **focusing on comparisons with other pose-free approaches.**
>
> **Additional Challenging Scenario Experiments:** To further demonstrate event camera advantages, we constructed a FASTER scenario by selecting every 10th frame from the SLOW dataset and directly applied COLMAP pose estimation on these challenging sequences.
> We added COLMAP+3DGS as a comparison baseline. Notably, in the corner and outdoor scenes, several frames exhibit registration failures where the estimated poses greatly deviate from the overall video trajectory, indicating COLMAP's limitations in challenging real-world scenarios.
>
> | Methods | SLOW | | FAST | |  FASTER | |
> |--|---|---|--|---|--|--|
> | | PSNR↑ | SSIM↑ | PSNR↑ | SSIM↑ |PSNR↑ | SSIM↑ |
> | CF-3DGS | 22.68 | 0.629 | 17.59 | 0.520 | 15.61 | 0.504 |
> | COLMAP+3DGS | 23.27 | 0.633 | 19.79 | 0.548 | 17.25 | 0.521 |
> | EF-3DGS (Ours) | **23.65** | **0.647** | **21.12** | **0.562** | **20.18** | **0.532** |
>
> These results demonstrate that: (1) COLMAP+3DGS rendering quality degrades significantly as motion speed increase confirming pose estimation inaccuracies in challenging conditions. (2) Compared to CF-3DGS, our method maintains substantially better rendering quality in high-speed scenarios, highlighting our method robustness and demonstrating the potential of event cameras in high-speed scenarios.
>
> We appreciate the reviewer's valuable suggestion and will include these challenging scenario comparisons in the revised manuscript to better highlight the effectiveness of our proposed event-aided 3DGS framework.
>
> > ### **Quality 3: Contrast Threshold Parameter Assumption**
>
> **Answer:** We appreciate the reviewer's concern about the contrast threshold assumption. To address this limitation, we propose a feasible approach that eliminates the dependency on the contrast threshold parameter through normalization, similar to scale-invariant processing in depth estimation.
>
> Specifically, for $L_{EGM}$, we use the brightness change calculation method, letting $\Delta \hat{L} = \hat{L}\_t - L\_{i,0}$, and $\Delta L = \sum\_{i=1}^{n} E\_{i,n}\cdot C$, then formulate a normalized $L_{EGM}$,
> $$
> L\_{EGM}= ||\frac{\Delta \hat{L}}{||\Delta \hat{L}||\_2} - \frac{\Delta L}{||\Delta L||\_2}||\_2.
> $$
> For $L_{LEGM}$, we apply similar normalization. We conducted experiments to validate this approach:
>
> | | PSNR↑ | SSIM↑   | $RPE_t$ ↓ | $RPE_r$ ↓ |
> |-|-|-|-|-|
> | normalization | 23.55 | 0.71  | 0.054 | 0.718 |
> | predifined $C$ (Ours) | **23.96** | **0.72**  | **0.049** | **0.626** |
>
> The results show that while the normalization eliminates threshold dependency and maintains reasonable performance, the predefined threshold method achieves better overall reconstruction quality and pose accuracy. This suggests that an accurate pre-computed threshold provides superior optimization constraints for our framework.
>
> > ### **Question 1: IPWE vs IWE Computational Cost**
>
> **Answer:** We conducted experiments comparing our IPWE approach with the individual event warping method from SSL-E2VID[3] to quantify the computational efficiency improvement. The experimental results and training times are shown below:
>
> |  | PSNR↑ | SSIM↑   | $RPE_t$ ↓ | $RPE_r$ ↓  | Training Time↓|
> |-|--|--|-|----|---|
> | IWE | 23.82 | **0.72**  | **0.048** | **0.603** | 7.3h |
> | IPWE  (Ours) | **23.96** | **0.72**  | 0.049 | 0.626 |**2.5h** |
>
> Our IPWE approach achieves a significant reduction in training time while maintaining comparable reconstruction quality. The individual event warping (IWE) shows a significant computational overhead. This is primarily due to: (1) higher memory requirements for storing each event as (x,y,p,t) tuples, and (2) substantially increased per-iteration computation from warping individual event.
>
> > ### **Question 2 & 3 & 4: Some Clarifications**
>
> **Answer:** We appreciate the reviewer's questions and will clarify these points in the revised manuscript:
>
> **1) Training Stage Ratio:** Yes, the ratio refers to the proportion of training iterations between the two stages.
>
> **2) Ev-Baseline in Table:** Ev-Baseline refers to EvCF-3DGS. We renamed it to "EvCF-3DGS" for better method description clarity. This appears to be an oversight where we missed updating the table reference during writing.
>
> **3) Initial Camera Pose:** We follow a progressive optimization strategy where each new frame's pose is initialized using the pose from the previous frame. This provides a reasonable starting point based on the assumption of smooth camera motion between consecutive frames.
>
> [1] Pan, Liyuan, et al. "Bringing a blurry frame alive at high frame-rate with an event camera." Proceedings of the IEEE/CVF conference on computer vision and pattern recognition. 2019.
>
> [2] Cannici, Marco, and Davide Scaramuzza. "Mitigating motion blur in neural radiance fields with events and frames." Proceedings of the IEEE/CVF Conference on Computer Vision and Pattern Recognition. 2024.
>
> [3] Paredes-Vallés, Federico, and Guido CHE De Croon. "Back to event basics: Self-supervised learning of image reconstruction for event cameras via photometric constancy." Proceedings of the IEEE/CVF Conference on Computer Vision and Pattern Recognition. 2021.

---

> > ### Comment · Reviewer_UuWp · 2025-08-04
> >
> > Dear authors,
> >
> > Thank you for taking time to answer my concerns.
> > Integration of EDI model and eliminating dependence on event threshold through normalization are reasonable.
> > Furthermore, I believe that the finding that replacing IWE with IPWE can reduce training time while maintaining image quality provides valuable insight for other related methods that utilize Contrast Maximization framework.
> > Most of my concerns were adequately addressed in the rebuttal; therefore, I raise my rating.

---

> > > ### Author Response · Authors · 2025-08-04
> > > **Thanks to Reviewer UuWp**
> > >
> > > Dear Reviewer UuWp,
> > >
> > > Thank you for improving the score of our paper. We are glad that our additional experiments were helpful to address your valuable concerns and appreciate your recognition of our contributions regarding the IWE and IPWE comparison. Your insightful feedback not only guided us to strengthen our work but also helped us better explain the technical merits of our approach, and we truly appreciate your thoughtful consideration throughout the review process.
> > >
> > > We will incorporate all your suggestions and clarifications into the revised version of our manuscript.
> > > Thanks again for your feedback and for recognizing the contribution of our work.
> > >
> > > Best regards,
> > >
> > > Authors of Submission 21739

---

### Official Review · Reviewer_hsxP · 2025-07-01

**Clarity:** 4
**Significance:** 2
**Originality:** 3
**Rating:** 5
**Confidence:** 5

**Summary:**

This paper introduces the event cameras to aid scene construction from a casually captured video for the first time, and propose Event-Aided Free-Trajectory 3DGS, called EF-3DGS, which seamlessly integrates the advantages of event cameras into 3DGS through three key components

**Questions:**

Please see the weakness.

**Ethical Concerns:**

["NO or VERY MINOR ethics concerns only"]

**Final Justification:**

Authors have solved my problem and i raise my score to accept, but my main concern is that this two-stage optimization strategy is not elegant enough and i believe there will be potential solutions.

**Limitations:**

See the weakness.

**Quality:**

3

**Strengths And Weaknesses:**

Strengths

Clarity and Visualizations: The paper is exceptionally well-written, and the figures are clear and easy to understand.

Rich Content: The overall content presented in the paper is comprehensive and substantial.

Weaknesses:

1. Necessity of Two-Stage Optimization: The paper proposes a two-stage framework: first optimizing the shape of Gaussian spheres for texture features, and then optimizing colors. While this idea is novel, its necessity is debatable. Specifically, in Section 4.1 (EGM optimization), could the optimization directly use the given reference image plus event data? For example, by calculating the difference between the 3DGS-rendered image and the reference image, and then using this difference with event data as a loss function. Combined with the authors' introduced optical flow warping, could this directly address the color issue? I would be very interested in seeing an ablation study on this specific aspect.

2. "Pose-Free" Claim Discrepancy: The paper's title claims a "pose-free" approach, yet the methodology section lacks any explicit description or discussion of how this pose-free aspect is achieved. Is this work directly built upon an existing pose-free 3DGS method, or is there a novel contribution to pose estimation that is not clearly articulated?

3. Justification for Pose-Free Approach: The necessity of a pose-free method needs stronger emphasis. A detailed discussion on how inaccuracies in pose estimation in real-world scenarios significantly impact other methods would strengthen the paper's claims and highlight the importance of their "pose-free" contribution.

---

> ### Author Rebuttal · Authors · 2025-07-31
>
> We thank the reviewer for the constructive feedback. We appreciate the recognition of our paper's writing and comprehensive content. We respond to each concern in detail below, and welcome any additional questions during the discussion period.
>
> > ### **W1: Necessity of Two-Stage Optimization**
>
> **Answer:** We acknowledge that our method description may have caused some confusion, and we would like to address this concern from two perspectives: 1) clarification of our current approach and 2) exploration of the alternative solution suggested by the reviewer.
>
> **1) Clarification of EGM Mathematical Equivalence**: The reviewer's proposed solution is actually mathematically equivalent to our EGM formulation. The suggested loss function can be expressed as:
> $$\min~||\frac{\hat{I}\_t}{I\_{i,0}} - \exp{(\sum\_{i=1}^{n} E\_{i,n}\cdot C)}||,$$
>
> where $\frac{\hat{I}\_t}{I\_{i,0}}$ represents the difference between rendered and reference images, and $\exp{(\sum\_{i=1}^{n} E\_{i,n}\cdot C)}$ corresponds to the brightness changes recorded by event data. By multiplying both sides by $I\_{i,0}$, the loss function becomes:
> $$||\hat{I}\_t - I\_{i,0}\cdot\exp{(\sum\_{i=1}^{n} E\_{i,n}\cdot C)}||$$
> According to our Eq. 5, $I\_t = I\_{i,j} = I\_{i,0}\cdot\exp{(\sum\_{i=1}^{n} E\_{i,n}\cdot C)}$, which transforms the objective to
> $$||\hat{I}\_t - I\_t||=L\_1(\hat{I}\_t, I\_t).$$
> This demonstrates the mathematical equivalence between the reviewer's suggestion and our EGM formulation.
> $$\min||\frac{\hat{I}\_t}{I\_{i,0}} - \exp{(\sum\_{i=1}^{n} E\_{i,n}\cdot C)}||\Leftrightarrow\min||\hat{I}\_t - I\_{i,0}\cdot\exp{(\sum\_{i=1}^{n} E\_{i,n}\cdot C)}|| \Leftrightarrow \min||\hat{I}\_t - I\_t||.$$
>
> Since event data lacks color information, the loss computation must be performed in grayscale space, resulting in gradients that also lack color information. Therefore, the reviewer's suggested formulation cannot address the color distortion problem.
>
> **2) Exploration of Alternative Approach**: Although the reviewer's proposed approach is mathematically equivalent to our EGM, inspired by the reviewer's constructive feedback, we investigated whether adding a reference view rendering loss could address this challenge:
> $$L\_{color} = (1 - \lambda)L\_1(\hat{I}\_{i,0}, I\_{i,0}) + \lambda L\_{D-SSIM}(\hat{I}\_{i,0}, I\_{i,0}).$$
> Given that events have much higher temporal resolution than RGB frames, the sparse color constraints from RGB images are largely overwhelmed by the abundant grayscale constraints from event data, ultimately resulting in color distortion. By incorporating $L\_{color}$, the backpropagated gradients at each iteration would be augmented with color information from the reference view, potentially alleviating the color distortion issue. We conducted additional experiments comparing this approach with our two-stage strategy:
> | Method Description | PSNR↑ | SSIM↑ | LPIPS↓ | Training Time↓
> |------------|-----|-----|-----|------|
> | $L\_{event}$   | 23.09| 0.70| 0.38|2.7h|
> | $L\_{event} + L\_{color}$   | **24.11** | **0.73** | **0.36** |3.8h|
> | $L\_{event}$ + two stage(Fixed GS) strategy   | 23.96| 0.72| **0.36**|**2.5h**|
>
> The results show that incorporating $L\_{color}$​ achieves comparable rendering quality.
> Although we cannot include visualizations in this rebuttal, our qualitative results confirm that the color distortion issue is effectively alleviated with this approach.
> However, this comes at a significant computational cost, increasing training time (3.8h vs 2.5h) due to additional reference view rendering at each iteration. Considering the modest performance difference relative to the substantial training overhead, we believe the two-stage optimization strategy offers a more efficient and practical solution.
>
> **Summary:** We appreciate the reviewer's insights. We proved the mathematical equivalence between the reviewer's suggestion and our EGM formulation. Inspired by the reviewer's feedback, we explored a viable alternative approach. However, our experiments demonstrate that our two-stage strategy is efficient and effective.
>
> > ### **W2: "Pose-Free" Claim Discrepancy**
>
> **Answer:** We thank the reviewer for this important clarification request. We sincerely apologize for not clearly explaining our pose-free implementation in the methodology section, as we focused primarily on introducing the event camera integration while overlooking the description of our overall pose-free pipeline. We address this concern in two parts:
>
> **1) How Pose-Free is Achieved**: Our pose-free approach is implemented through a progressive joint optimization framework detailed in Section 4.5 "Overall Training Pipeline" and Algorithm 1 (Appendix A.3). Specifically, we build upon the optimization scheme of CF-3DGS. As shown in Algorithm 1 in the appendix, we progressively optimize camera poses for each timestamp following the temporal order of video frames using our proposed event-driven loss $L_{event}$. During this process, we alternate between optimizing camera poses and reconstructing the 3DGS. Our two-stage Fixed-GS training strategy (Section 4.4) is integrated within the scene reconstruction step to mitigate color distortion issues. To handle long video sequences and OOM(out of memory) problem, we adopt LocalRF's local radiance field allocation strategy, allocating a new 3DGS every 50 frames.
>
> **2) Novel Contribution Clarification**: While we build upon CF-3DGS's optimization framework, our core contribution is not proposing an entirely new joint pose-scene optimization method. Instead, our novelty lies in effectively leveraging event camera properties to address two fundamental challenges that existing pose-free methods face in high-speed scenarios: (1) sparse scene reconstruction due to insufficient frame observations, and (2) inaccurate pose estimation caused by large inter-frame displacements. Our event-driven optimization provides dense supervision through EGM and extracts motion information via LEGM. The experiments across different frame rates (6FPS to 1FPS) validate that our performance advantage becomes greater as camera speed increases. Additionally, to address the color distortion issue introduced by event integration, we propose a simple but effective Fixed-GS two-stage optimization strategy that separates structure and color optimization.
>
> We will pay more attention to the method description, ensuring that the pose-free implementation is well explained and its significance is highlighted.
>
>
> > ### **W3: Justification for Pose-Free Approach**
>
> **Answer:** We thank the reviewer for this valuable feedback. Pose-free methods are indeed crucial because traditional 3DGS methods rely on COLMAP for pose estimation, which frequently fails in challenging scenarios such as low-texture regions and high-speed motion. Nope-NeRF has demonstrated this issue through examples of COLMAP failure cases [1], validating the necessity of pose-free approaches. In fact, pose-free NeRF and 3DGS represent a very active research direction, with continuous developments, including BARF, Nope-NeRF, LocalRF, and CF-3DGS.
>
> We apologize for not sufficiently emphasizing the critical issue of pose estimation degradation in high-speed scenarios. Our experiments demonstrate that existing pose-free methods suffer from severe pose estimation errors when camera motion becomes rapid. As shown in Table 2, on Tanks & Temples, as frame rate decreases from 6FPS to 1FPS, our method maintains significantly better pose accuracy (40% lower ATE) compared to CF-3DGS. Similarly, Table 3 shows superior robustness in FAST scenarios on RealEv-DAVIS. Our supplementary video provides visual evidence. The video clearly shows that CF-3DGS produces severe rendering artifacts (scene discontinuities) in high-speed scenarios due to inaccurate pose estimation, while our event-aided approach maintains stable pose estimation and high-quality rendering. Furthermore, we conducted supplementary experiments to validate the pose estimation inaccuracy issues of COLMAP and other pose-free methods in high-speed scenarios. Please refer to the "**Additional Challenging Scenario Experiments**" section in our response to **Quality 2** of **reviewer UuWp's** reviews.
>
> We will revise the paper and add a detailed discussion to better highlight this critical contribution and its significance in the context of existing pose-free methods.
>
> [1] Bian, Wenjing, et al. "Nope-nerf: Optimising neural radiance field with no pose prior." Proceedings of the IEEE/CVF Conference on Computer Vision and Pattern Recognition. 2023.

---

> > ### Comment · Reviewer_hsxP · 2025-08-06
> > **Thanks for solving my problems.**
> >
> > Thanks to the authors for solving my problem, i have rased my score to accept, but my main concern is that this two-stage optimization strategy is not elegant enough and i believe there will be potential solutions, looking forward to your further exploration.

---

> ### Author Response · Authors · 2025-08-06
> **Thanks to Reviewer hsxP**
>
> Dear Reviewer hsxP,
>
> Thank you very much for improving the score of our paper! We are deeply grateful for your constructive feedback, particularly your thoughtful insights regarding the two-stage optimization strategy. During our early exploration, we actually investigated alternative approaches, such as experimenting with event-guided gradients to preserve brightness influence while eliminating color impact, but these attempts did not yield satisfactory results, which led us to adopt the current two-stage strategy as a practical compromise solution. We totally agree that "there will be potential solutions" and look forward to discovering more elegant solutions in future work.
>
> We will incorporate all your suggestions into the revised manuscript. Once again, thank you for your time and constructive feedback : )
>
> Best regards,
>
> Authors of Submission 21739

---

### Official Review · Reviewer_cYjR · 2025-07-03

**Clarity:** 2
**Significance:** 3
**Originality:** 3
**Rating:** 5
**Confidence:** 2

**Summary:**

This paper proposed EF-3DGS. It is a 3DGS model based on 1. high-frequency event camera stream 2. RGB video frames and 3. no trajectories aka camera poses. The overall problem setup is compelling since event camera captures a continuous stream that encapsulate fine-grained motion and RGB video frames contains rich visual details for constraining the overall structure. Such a setup without any presumption on trajectory or poses is worth studying. This paper integrates the principle of EGM, CMax and LEGM into this setup. Further, the proposed PBA refines the continuous trajectory supervised by sparse video frames. The overall pipeline is split into two stage where the first stage optimize the overall geometry and monochrome color and second stage refines the color with RGB video stream. Results show effectiveness on both reconstruction quality and pose/trajectory estimation.

**Questions:**

Such setup is quite commonly observed on robotics and autonomous vehicles. I would like to know you thoughts on how far away from EF-3DGS to real-time, online processing, since inherently your work does not need any trajectory. Is it straight forward or still unanswered questions?

minor suggestions:

- place figure 3 and figure 6 in a better position, somewhere closer to it is first mentioned.
- some basic introduction around EGM, CMax and LEGM might go to the preliminary sections. Otherwise it is confusing which part in the methodology is the contribution of this work.
- It can also be interesting to visualize the optimization process, that as loss being optimized, the warped image gets sharper and trajectory is optimized simultaneously.

**Ethical Concerns:**

["NO or VERY MINOR ethics concerns only"]

**Final Justification:**

I remain the opinion that this paper is a qualified contribution to NeurIPS and scores are remained as well.

**Limitations:**

yes

**Quality:**

3

**Strengths And Weaknesses:**

Strength

- The  proposed pipeline which integrates different modalities, 3DGS, event generation model and bundle adjustment, is overall sound and reasonable.
- Sufficient experiments that both presents results quantitatively and qualitatively, with visualizations and numerical comparisons. Results show good performance on both reconstruction and trajectory recovery.

weakness

- Figure 3 might be too small can hardly tell much from the comparison (but the one in the appendix is ok so not a big issue). The location of figure 3 need adjust as well.
- With the loss mentioned in eq.18, both rgb and grayscale loss are involved. It is unclear how they are optimized together.

Methodology part need some reorganization:

- on page 5 where IPWE is introduced, it is illustrated in Figure 6 but Figure 6 is on page 9.
- in 4.4, the argmin equation is minimizing $\mathcal{L}_{event}$ but this loss is not introduced until 4.5.

---

> ### Author Rebuttal · Authors · 2025-07-31
>
> We thank the reviewer for the constructive feedback and for recognizing our compelling problem setup, sound pipeline, and sufficient experimental validation. We respond to each concern in detail. Please feel free to use the discussion period if you have any additional questions.
>
> > ### **W1 & W3 & W4 & Minor S1: Paper format and reorganization issues.**
>
> **Answer:** We apologize for the confusion caused by the figure size, figure placements and ordering issues in our methodology section. We will address these issues in the revision by adjusting the figure size, moving figures closer to where they are mentioned, and ensuring all concepts are defined before they're used. Thanks for bearing with these organization issues.
>
> > ### **W2: Confusion about loss optimization in Eq. 18.**
>
> **Answer:** We clarify that RGB and grayscale losses can be optimized simultaneously. For grayscale losses ($L_{EGM}$ + $L_{LEGM}$), we convert 3DGS RGB renderings to grayscale using $Y = 0.299 \cdot R + 0.587 \cdot G + 0.114 \cdot B$, then compute the losses. The RGB loss ($L_{PBA}$) is computed directly on the original RGB renderings. For conciseness in our mathematical formulation, we did not explicitly indicate the color space for each loss computation. We will clarify this in the revised version.
>
> > ### **Q1: Potential of EF-3DGS for real-time and online applications.**
>
> **Answer:**
>
> ​	Thanks for your interest in the real-time feasibility of our approach, which is indeed crucial for robotics and autonomous driving applications. We respond from two aspects: 1) Current performance analysis and 2) future directions.
>
> ​	**1) Current performance analysis**: We have provided a comprehensive computational analysis in Appendix A.7. For rendering speed, our method achieves over 30 FPS, approaching real-time performance. For training time, our approach achieves significantly better reconstruction quality than previous pose-free methods (CF-3DGS, LocalRF) without substantially increasing computational overhead. However, we acknowledge that all current pose-free methods, including ours, require per-scene optimization, making real-time reconstruction still challenging.
>
> ​	**2) Future directions**: Recent work VGGT[1] demonstrates promising directions by utilizing large Transformers to regress reconstructed point clouds and poses within seconds from input images. We believe combining the VGGT paradigm with EF-3DGS presents tremendous opportunities, though several challenges remain: 1) How to extend point cloud reconstruction to 3DGS reconstruction. 2) How to enhance model performance in high-speed (sparse view) scenarios. 3) How to train a unified model receiving both events and images as inputs, even if they have significant data modality differences.
>
> ​	We are excited about these directions and believe they represent the next frontier for real-time and pose-free scene reconstruction. As you noted, these remain challenging questions. However, we view them as promising avenues for future research.
>
> > ### **Minor S2: Methodology organization of EGM, CMax, and LEGM.**
>
> **Answer:** Thank you for your suggestion. We agree that a better organization would significantly improve clarity and better distinguish our contributions. We will reorganize the methodology section for better clarity. We plan to move EGM part to the preliminary section as it describes the fundamental working principle of event cameras. CMax and LEGM parts will remain in the methodology section as they represent the core components of our approach: extracting motion information from event streams for pose optimization, and providing gradient-domain constraints through warped events for well-constrained scene reconstruction.
>
> > ### **Minor S3: Visualization of the optimization process.**
>
> **Answer:** Thanks for your suggestion. We have actually provided trajectory estimation visualization in our supplementary materials (videos), which shows the pose estimation process as video frames and event streams are progressively input. However, due to current rebuttal format constraints, we plan to include the specific visualization of warped events during optimization in the future revised version.
>
>
>
> [1] Wang, Jianyuan, et al. "Vggt: Visual geometry grounded transformer." *Proceedings of the Computer Vision and Pattern Recognition Conference*. 2025.

---

### Note · Authors · 2025-08-12

We sincerely thank all reviewers for their thoughtful evaluation and constructive feedback. We are encouraged that the reviewers appreciate our work, including:

- Our approach effectively achieves pose-free reconstruction by leveraging complementary strengths of event and RGB cameras, which reviewers found "sound and reasonable" for high-speed scenarios **[cYjR]**.

- Multiple reviewers praised our clear writing and presentation, noting the paper is "well-written" **[hsxP, rd3s]** with figures that are "easy to understand" **[hsxP]**.

- The comprehensive experimental validation and extensive quantitative and qualitative experimental results **[cYjR, hsxP, rd3s]**.

Each reviewer also provided constructive feedback that helped improve our work.

**Shared concerns:** Multiple reviewers raised questions about the pose-free aspects - its description, experimental setting, and methodology. We addressed these through detailed clarifications and explanations. Additionally, we conducted further experiments in FASTER scenarios (where COLMAP fails), demonstrating the advantages of our method under challenging scenarios.

We are pleased that our responses have helped clarify these concerns. We appreciate the reviewers' guidance in helping us improve the clarity. We will incorporate these improvements into the final version.

**Individual concerns:**

For reviewer **cYjR**: We reorganized the paper following the suggestions and explored promising future research directions for real-time applications.

For reviewer **hsxP**: We proved the mathematical equivalence between the suggested approach and our EGM formulation. Inspired by the suggestion, we explored an alternative solution, confirming the computational efficiency of the two-stage optimization strategy.

For reviewer **UuWp**: We clarified our focus on inter-frame challenges and extended our method with the EDI model to handle motion blur. We explored threshold-free approaches through normalization and compared the efficiency between IWE and IPWE.


For reviewer **rd3s**: We provided theoretical analysis and noise experiments demonstrating our method's robustness. We shared our thoughts on potential robustness in $L_{grad}$ that might help deepen the understanding of our method.

We are grateful that all reviewers responded favorably to our rebuttal, with multiple reviewers raising their scores. We will incorporate all suggested improvements into the final manuscript. Thank you for your consideration.

---

### Decision · Program_Chairs · 2025-09-17

**Decision:**

Accept (spotlight)

**Comment:**

This paper introduces EF-3DGS, a novel framework that integrates event cameras with 3D Gaussian Splatting for pose-free scene reconstruction. The method is well-motivated, clearly presented, and extensively validated across synthetic and real datasets, showing notable gains in both pose estimation and rendering quality, especially under high-speed scenarios. Reviewers praised the clarity, soundness of the pipeline, and thorough experiments, while raising concerns about the elegance of the two-stage optimization, assumptions in experimental settings (e.g., motion blur, thresholding), and the need for more discussion of limitations. The rebuttal was detailed, provided additional experiments (including blur robustness and challenging scenarios), and addressed most concerns satisfactorily. Overall, this is a solid contribution, and I recommend acceptance.